



# Oxidation product characterization from ozonolysis of the diterpene ent-kaurene

Yuanyuan Luo[1], Olga Garmash[1,2], Haiyan Li[1,3], Frans Graeffe[1], Arnaud P. Praplan[4], Anssi Liikanen[4], Yanjun Zhang[1,5], Melissa Meder[1], Otso Peräkylä[1], Josep Peñuelas[6,7], Ana María Yáñez-Serrano[6,7,8], Mikael Ehn[1]

[1]Institute for Atmospheric and Earth System Research/Physics, Faculty of Science, University of Helsinki, Helsinki, 00014, Finland
[2]Aerosol Physics Laboratory, Physics Unit, Tampere University, Tampere, 33014, Finland
[3]School of Civil and Environmental Engineering, Harbin Institute of Technology, Shenzhen, 518055, China
[4]Atmospheric Composition Research, Finnish Meteorological Institute, Helsinki, 00101, Finland
[5]Univ Lyon, Université Claude Bernard Lyon 1, CNRS, IRCELYON, Villeurbanne, 69626, France
[6]CREAF, Bellaterra (Cerdanyola del Vallès), Catalonia, E08193, Spain
[7]CSIC, Global Ecology Unit, CREAF-CSIC-UAB, Bellaterra (Cerdanyola del Vallès), Catalonia, E08193, Spain
[8]IDAEA-CSIC, Barcelona, 08034, Spain

*Correspondence to*: Yuanyuan Luo (yuanyuan.luo@helsinki.fi) and Mikael Ehn (mikael.ehn@helsinki.fi)

**Abstract.** Diterpenes ($C_{20}H_{32}$) are biogenically emitted volatile compounds that only recently have been observed in ambient air. They are expected to be highly reactive, and their oxidation is likely to form condensable vapors. However, until now, no studies have investigated gas-phase diterpene oxidation. In this paper, we explored the ozonolysis of a diterpene, ent-kaurene, in a simulation chamber. Using state-of-the-art mass spectrometry, we characterized diterpene oxidation products for the first time, and we identified several products with varying oxidation levels, including highly oxygenated organic molecules (HOM) monomers and dimers. The most abundant monomers measured using a nitrate chemical ionization mass spectrometer were $C_{19}H_{28}O_8$ and $C_{20}H_{30}O_5$, and dimers were $C_{38}H_{60}O_6$ and $C_{39}H_{62}O_6$. The exact molar yield of HOM from kaurene ozonolysis was hard to quantify due to uncertainties in both the kaurene and HOM concentrations, but our best estimate was a few percent, which is similar to values reported earlier for many monoterpenes. We also monitored the decrease of the gas-phase oxidation products in response to an increased condensation sink in the chamber to deduce their affinity to condense. The oxygen content was a critical parameter affecting the volatility of products, with 4–5 O-atoms needed for the main monomeric species to condense. Finally, we report on the observed fragmentation and clustering patterns of kaurene in a Vocus proton transfer reaction time-of-flight mass spectrometer. Our findings highlight similarities and differences between diterpenes and smaller terpenes during their atmospheric oxidation, but more studies on different diterpenes are needed for a broader view of their role in atmospheric chemistry.





## 1 Introduction

Terpenes consist of isoprene ($C_5H_8$), monoterpenes ($C_{10}H_{16}$), sesquiterpenes ($C_{15}H_{24}$), diterpenes ($C_{20}H_{32}$), triterpenes ($C_{30}H_{48}$), and even more complex compounds. The smaller terpenes (isoprene, monoterpenes, and sesquiterpenes) are the most abundant

biogenic volatile organic compounds (BVOCs) in the atmosphere (Guenther et al., 1995), estimated to account for 40–65% of the ~1000 Tg yr$^{-1}$ of BVOCs mass released by vegetation (Arneth et al., 2008; Acosta Navarro et al., 2014; Guenther et al., 2012). Typically, with one or more double bonds, terpenes display a high diversity in structures and reactivities. Once emitted into the atmosphere, terpenes play an important role in atmospheric chemistry. They can undergo various oxidation reactions with different atmospheric oxidants, such as ozone ($O_3$), hydroxyl radical (OH), and nitrate radical ($NO_3$), forming a wide

range of new oxygenated organic species. These reactions impact pollutants like tropospheric $O_3$ and organic aerosol particles (Atkinson and Arey, 2003; Calogirou et al., 1999). In addition to their detrimental health impacts, aerosol particles also influence Earth's radiation budget and, consequently, climate (IPCC, 2021; Ezhova et al., 2018; Pöschl, 2005).

BVOCs oxidation products can contribute to atmospheric aerosol loadings if their vapor pressures are low enough. Over the past decades, extensive studies have been conducted to investigate the oxidation mechanism and the role of terpenes in

secondary organic aerosols (SOA) formation around the world (Mohr et al., 2017; Zhang et al., 2018; Li et al., 2021; Kontkanen et al., 2016; Ehn et al., 2012; Yassaa et al., 2012), in particular from isoprene and monoterpenes. Isoprene has a relatively low yield of SOA formation, while monoterpenes play a more vital role in the SOA formation with typical SOA yields of ~5–10% (Mutzel et al., 2016; Griffin et al., 1999; Mcfiggans et al., 2019; Zhang et al., 2018; Ehn et al., 2014). Also, sesquiterpenes have recently received growing interest and have been reported to be even more efficient SOA contributors, with reported

SOA yields from 6% up to 100% (Jaoui et al., 2013; Lee et al., 2006; Boy et al., 2007; Li et al., 2011). However, with the yearly emissions of isoprene, monoterpenes, and sesquiterpenes being around 500 Tg yr$^{-1}$, 160 Tg yr$^{-1}$, and 30 Tg yr$^{-1}$ (Guenther et al., 2012), respectively, the relative contributions of these different compound groups to SOA are not easy to determine. This is even more true for other terpene groups, for which very few studies exist to date.

Diterpenes have previously been identified in some extractions of trees such as leaf oils, pine needles, and resins (Kato, 2005;

Keeling and Bohlmann, 2006; Lee et al., 2009; Tumen et al., 2010); however, they were for a long time thought not to be emitted into the atmosphere due to their fairly low volatility (Guenther, 2002). Matsunaga et al. (2012) were the first to report diterpene kaurene emissions from branch enclosure experiments of coniferous trees in Japan. After that, several studies reported the observation of diterpenes using similar branch enclosure systems with large variation in the estimated emission rates in different studies (Haberstroh et al., 2018; Lin et al., 2015; Yáñez-Serrano et al., 2018; Helin et al., 2020). For instance,

the estimated emission rates of diterpenes were 1–3 orders of magnitude lower than that of monoterpenes and sesquiterpenes in the boreal forests (Helin et al., 2020). However, in a study of Mediterranean shrubs and a study in Japan during warm seasons (temperature ~ 30 ℃), the emission rates for diterpenes were comparable to those of monoterpenes (Matsunaga et al., 2012; Yáñez-Serrano et al., 2018). Additionally, diterpene emission rates could be further increased under stress conditions like heat and mechanical wounding (Lin et al., 2015).



Very recently, direct ambient observations of diterpenes have become feasible with the latest developments of mass spectrometric techniques. To our knowledge, three studies have reported diterpenes in the atmosphere, but diterpene concentrations were much lower compared to monoterpenes and sesquiterpenes (Chan et al., 2016; Li et al., 2020; Yee et al., 2018). For example, Li et al. (2020) directly measured around 1.7 ppt (part per trillion) of diterpenes in the French Landes forest during summertime, 10–1000 times lower than monoterpenes and sesquiterpenes. Moreover, four diterpenes (the most

abundant one was kaurene) with concentrations ranging from 10 to 86 ppq (part per quadrillion) were detected at a rural site in the Amazon region by Yee et al. (2018), while most detected sesquiterpenes were above 100 ppq. The low concentrations are likely primarily linked to the lower emissions rates of the less volatile diterpenes compared to smaller terpenes, and their reactivities are also likely to be high once in the atmosphere. Although no diterpene oxidation products have been reported, it can be assumed that oxidation is a main sink of diterpenes, with unknown impacts on atmospheric chemistry and SOA

formation.

In this study, we characterize the oxidation products formed via ozonolysis of the diterpene ent-kaurene (hereafter referred to as kaurene). The precursor and oxidation products in the gas phase were measured using state-of-the-art mass spectrometers (Jokinen et al., 2012; Krechmer et al., 2018). We present the first comprehensive summary of the identified oxidation products with different oxidation levels, including highly oxygenated organic molecules (HOM, (Ehn et al., 2014; Bianchi et al., 2019)).

We also probe the potential of the oxidation products to condense onto existing particles to form SOA. Finally, we discuss fragmentation and clustering patterns of kaurene using the Vocus proton transfer reaction time-of-flight (PTR-TOF) mass spectrometer.

## 2 Methods

### 2.1 Chamber set-up

Kaurene ozonolysis experiments were conducted in a 2 m$^3$ Teflon (FEP, supplied by Vector Foiltec, Germany) chamber, the COALA chamber at the University of Helsinki, Finland. Further details of the chamber facility can be found elsewhere (Riva et al., 2019b; Peräkylä et al., 2020). During the experiments, the chamber was operated under steady-state conditions with a continuous inflow of ~36 L min$^{-1}$, and the residence time was around 56 min. The injected flow consisted of purified air generated by a clean air system (AADCO, Series 737-14, Ohio, USA), with additions of varying amounts of O$_3$ (Dasibi 1008-

PC ozone generator) and kaurene, as well as 80 nm ammonium sulfate (AS) seed particles. Instruments sampled the majority of the outflow for chemical composition measurement both in the gas and the particle phase (Sect. 2.2), while the rest was flushed into an exhaust line. O$_3$ concentration in the chamber was monitored by a UV photometric analyzer (Model 49P, Thermo-Environmental) during the experiments. Temperature, relative humidity (RH), and pressure in the chamber were monitored using a Vaisala temperature and humidity probe (INTERCAP® HMP60) and a differential pressure sensor

(Sensirion SDP1000-L025). All experiments were carried out at room temperature (26 ± 1℃) under a slight overpressure condition to minimize leaks into the chamber, and with the RH consistently below 1%.





All instruments sampled the chamber air continuously, and the input into the chamber was varied to achieve different concentrations and ratios of kaurene and $O_3$. In addition, seed aerosols were added at different times in order to make condensation onto aerosols, rather than chamber walls, the dominant sink for low-volatile vapors. The injection of kaurene

was one of the most challenging parts of the experiments due to its low volatility. Kaurene is a solid at room temperature, and the kaurene in this study was purchased from OIChemIm s.r.o. Gas-phase kaurene was introduced to the chamber by flushing nitrogen ($N_2$) through a vial containing the solid kaurene, and the flow was subsequently directed into the chamber. In addition, a heater was placed under the vial, heating the air around to promote the evaporation of kaurene. During the experiments, the vial's bottom temperature stayed below 45°C, except for the last experiment day when the vial reached a temperature around

60°C. Already at small heating, the solid kaurene had melted to liquid form, but we did not see indications of decomposition of the kaurene from the heating in any of our mass spectra. Variable levels of kaurene in the chamber were achieved, either by turning the heater on/off or changing the flow rate of the $N_2$ carrier gas, which was controlled with a mass flow controller (MKS, G-Series, Andover, MA, USA). Due to the low volatility of kaurene, residual amounts were introduced to the chamber also when no active addition was performed, presumably due to evaporation from surfaces in the tubing and chamber.

**2.2 Instrumentation**

We deployed a suite of online instrumentation measuring gas and particle phase species. The Vocus PTR-TOF (Tofwerk AG/Aerodyne Research, Inc.) (Krechmer et al., 2018) was applied to determine the reactant kaurene concentration and some of its oxidation products. The instrument allows fast and continuous measurements of VOCs with sub-ppt detection limits (Krechmer et al., 2018). Based on the previous evaluation, the Vocus can also detect various oxidation products of

monoterpenes containing up to around six oxygen atoms (Riva et al., 2019). In this study, the instrument was run with the axial and radial voltages being 350 and 400 V, respectively. The pressure in the ionization regions (drift tube) was set to 1.4 mbar, giving an electric field strength E/N of 120 Td. However, as discussed in more detail in Sect. 2.3, the pressure drifted during the experiments, resulting in significant changes in the absolute sensitivity of the sampled molecules. Around 4.5 L min$^{-1}$ of the chamber air was directed into the Vocus through 1 m long PTFE tubing (6 mm o.d., 4 mm i.d.) for further analysis. A water

flow (HPLC-grade or 18-MOhm milli-Q water) at the rate of 15 sccm was utilized to produce reagent ions in a discharged ion source. The mass resolving power of the long TOF mass analyzer was 12 000–13 000, and data were recorded with a time resolution of 5 s.

For measuring HOM and other more oxygenated products from kaurene ozonolysis, a chemical ionization atmospheric pressure interface time-of-flight mass spectrometer (CI-APi-TOF, Tofwerk AG/Aerodyne Research, Inc.) (Jokinen et al., 2012)

was used. We used nitric acid ($HNO_3$) as the reagent and an X-ray source to produce nitrate reagent ions ($NO_3^-$). The sample molecules could be charged by collision with nitrate ion clusters ($NO_3^-$, $HNO_3NO_3^-$, and $(HNO_3)_2NO_3^-$). After a collision, the sample molecules were either ionized via a direct proton transfer and detected as a deprotonated ion or, more typically, via clustering with reagent ions and detected as adducts (Jokinen et al., 2012). During the experiments, the instrument, equipped





with a long TOF mass analyzer (mass resolving power 13 000–14 000), was configured to measure ions up to 960 Th with a
time resolution of 10 s. The sampling flow rate was around 10 L min⁻¹ for all experiments.

A custom-built scanning mobility particle sizer (SMPS) and a Long Time of Flight Aerosol Mass Spectrometer (L-ToF-AMS,
hereafter AMS, Aerodyne Research Inc.) were deployed to measure the particle phase. The L-ToF-AMS is similar to the High-
Resolution ToF-AMS (HR-ToF-AMS) described in Decarlo et al. (2006) but has a longer ToF-chamber for increased mass
resolving power.

Data of both the Vocus and the nitrate CI-APi-TOF was analyzed with a MATLAB tofTools package (version 607) (Junninen,
2014), while the ToF-AMS Analysis software packages SQUIRREL (version 1.63H) and PIKA (version 1.23H) (Sueper et al.,
2011) within Igor Pro (version 6.37, WaveMetrics Inc.) was used for the AMS data. The SMPS data were analyzed with
MATLAB and Igor Pro.

A system for total $O_3$ reactivity measurements (TORM) was also deployed during the experiments. The TORM consists of a
reactor (three 2 L borosilicate glass bottles) in which sampled air reacts with a set concentration of $O_3$ (150 ppb in this study)
produced by an $O_3$ generator. A modified $O_3$ analyzer (Model 49i, Thermo Scientific, Waltham MA, USA) operating in
differential mode recorded directly the difference in $O_3$ before and after the reactor, which is used to calculate the total $O_3$
reactivity in the COALA chamber. The kaurene concentration could be estimated based on the calculated $O_3$ reactivity data at
a given rate coefficient of kaurene-$O_3$ reaction, with the assumption that kaurene was the dominant sink for $O_3$. A detailed
explanation of the instrument and calculation can be found elsewhere (Helmig et al., 2021). During the experiments, the total
flow through the reactor was 6.0 L min⁻¹ with 5.0 L min⁻¹ sampling flow from the COALA chamber and 1.0 L min⁻¹ coming
from the $O_3$ generator (dilution factor f= 0.83). The residence time in the reactor was estimated to be 60 s.

Multiphase adsorbent (Tenax TA/Carbopack B, 60–80 mesh) tubes (o.d. 1/4 in × 3 1/2 in, PerkinElmer Inc., Waltham, MA,
USA) were utilized as one way to quantify diterpene concentration in the chamber due to the lack of any authentic diterpene
standards for Vocus. We collected four Tenax tubes during the experiments, and details about the sampling parameters can be
found in Table S1. After sampling, a thermal desorption-gas chromatograph (TD-GC, TurboMatrix 350 automatic TD unit,
Clarus 680 GC, PerkinElmer Inc.) coupled with a quadrupole mass spectrometer (MS, Clarus SQ 8 T, PerkinElmer Inc.) was
applied to determine and quantify kaurene in the Tenax samples. More details about the methodology and Tenax tubes have
been reported by Helin et al. (2020).

**2.3 Kaurene quantification and correction of Vocus time series**

In Vocus, the sampled air is charged in a focusing ion-molecule reactor (FIMR) by combining with the reagent ions produced
from a low-pressure discharge reagent ion source. The FIMR is a glass tube with four quadrupole rods installed outside in a
radial pattern. The FIMR pressure is controlled by a valve between the reactor and a mechanical pump. The FIMR pressure
typically remains constant due to the control where additional pumping is applied to it to keep a set pressure. However, in our
study, the pinhole through which the sample air entered the FIMR became partially blocked by seed aerosol particles. When
sufficiently blocked, the pressure control could no longer be maintained since even without additional FIMR pumping, the





pressure dropped below the desired 1.4 mbar, as the flow through the pinhole had decreased. As the pinhole clogging was not noticed during the experiments, the FIMR pressures kept decreasing in a stepwise pattern from the initial ~1.4 mbar at the beginning of the experiments to ~ 0.9 mbar in the end. The changes in FIMR pressure are expected to impact both the sensitivity

and the fragmentation behavior of the Vocus by affecting the frequency and energy of ion-molecule collisions. Hence, the raw signal intensity of kaurene could not directly depict the actual changes of kaurene in the chamber over the course of the experiments. Therefore, we needed to correct the sensitivity changes as far as possible to simulate the actual variation of kaurene using a simple model. For this purpose, we utilized the observed changes in chamber $O_3$ during kaurene additions. The details are described in the following.

The $O_3$ concentration was monitored online for all experiments. When we injected kaurene into an $O_3$-stable chamber, the $O_3$ loss rate increased due to reactions with kaurene (in addition to ever-present flush out). For any time point, we can write the change in $O_3$ concentration in the chamber as:

$$\frac{d[O_3]}{dt} = Q_{O3, \text{ in}} - [O_3] * [\text{kaurene}] * k_1 - [O_3] * k_{\text{flush-out}} \tag{1}$$

Where $Q_{O3, \text{ in}}$ is the injection rate of $O_3$, $[\text{kaurene}]$ and $[O_3]$ are the current kaurene and $O_3$ concentrations, respectively, $k_{\text{flush-out}}$ is the rate at which air is flushed out of the chamber, and $k_1$ is the kaurene-$O_3$ reaction rate coefficient.

For every experiment starting with a stable $O_3$ condition, we simulated a kaurene time series. We solve Eq. (1) for $[\text{kaurene}]$, and take it as the expected kaurene concentration at time i (kau_exp$_i$). $\frac{d[O_3]}{dt}$ was estimated based on the average change of $O_3$ from time point (i-1) to the time point (i+1) (time resolution: 15 min = 900 s), giving the expression finally as:

$$\text{kau\_exp}_i = \frac{Q_{O3,in} - \dfrac{O_{3,i-1} - O_{3,i+1}}{2 * 900 \text{ s}} - O_{3,i} * k_{\text{flush-out}}}{O_{3,i} * k_1} \tag{2}$$

The average residence time in the COALA chamber concerning flush-out was 56 min, and thus the reciprocal $k_{\text{flush-out}} = 3\times10^{-4}$ $s^{-1}$. $Q_{O3,in}$ (unit $cm^{-3}s^{-1}$) was estimated for each experiment from the steady-state periods before kaurene injection, when $\frac{d[O_3]}{dt} = 0$. The kaurene-$O_3$ reaction rate coefficient $k_1$ was set to be $5\times10^{-16} cm^3 molec^{-1}s^{-1}$ based on the model results of the rate coefficient tests of kaurene and $O_3$ performed in the beginning of our experiments (see Supplementary Material). This value is fairly close

to the only value $(1.4 \pm 8.2 \times10^{-15} cm^3 molec^{-1}s^{-1})$ so far reported in the literature (Helin et al., 2020).

The simulated kaurene concentrations were only reliable when kaurene concentrations were high enough (> ~1 ppb) to make a noticeable impact on the $O_3$ concentrations. We used these periods to scale the measured Vocus raw kaurene signal (kau_sig$_i$) to the simulated values (kau_exp$_i$) by a correction factor, cor_fac$_i$:

$$\text{cor\_fac}_i = \frac{\text{kau\_exp}_i}{\text{kau\_sig}_i} \tag{3}$$



We further assumed that this factor stayed constant as long as the pressure in the FIMR did not change, and thus extrapolated the correction factors also to periods with low kaurene concentrations. In this way, a complete and consecutive time series of correction factors for kaurene was achieved (see the blue line in Fig. 1(a)).

All the above corrections bring with them considerable uncertainty and potential error sources. In order to assess the reliability of the correction method, we compared our acquired kaurene time series with those measured by TORM and the four collected

Tenax samples. First, quantified kaurene concentrations by Tenax tubes are shown in Table S1. The average Vocus signal intensity of kaurene during the sampling time for each Tenax tube was calculated, and the corresponding correction factors required by the Vocus to match the Tenax concentrations can be found in Fig. 1(a). The correction factors we derived above were in the same order of magnitude as those deduced from the Tenax samples, although the Tenax samples consistently suggested lower kaurene concentrations. The third Tenax sample (Table S1) on January 28th was, however, two orders of

magnitude lower, which leads to expect that this sample was unsuccessful for unknown reasons. For comparison with TORM, we scaled the raw Vocus signal intensity data of kaurene to match the estimated kaurene based on TORM. The required scaling factor time series is shown in Fig. 1(a) for periods when the kaurene was above the detection limit (~1 ppb) of TORM. Overall, the TORM calibration factors compared well with those from $O_3$ depletion method above. The good agreement becomes evident when plotting the final kaurene time series from TORM and the corrected Vocus data (Fig. 1(b)), lending confidence

to the Vocus correction method. The figure also shows the importance of utilizing the Vocus data, as the TORM was not able to quantify kaurene concentrations below ~1 ppb due to background. Despite this, the correlation between the instruments was quite good ($R^2$=0.87, Fig. 1(c)).

Based on the comparison of our simulated kaurene results with these two independent methods, we conclude that our correction approach was successful. However, the uncertainties remain high due to the needed corrections, the lack of authentic standards,

and the dependence on needing to know the kaurene reaction rate coefficient with $O_3$. We estimate that the absolute concentrations have an uncertainty of at least a factor 3 (+200%/-67%), while the uncertainty in the relative changes over the course of our study is closer to a factor 2 (+100%/-50%).

## 2.4 HOM formation and quantification

### 2.4.1 HOM formation

Nearly all atmospheric VOCs oxidation will produce peroxyl radical ($RO_2$) intermediates, and the final distribution of oxidation products is largely determined by the reaction pathways available to these $RO_2$ (Atkinson and Arey, 2003). Autoxidation is one important propagation process for $RO_2$ recently shown to be of significance in the atmosphere (Crounse et al., 2013; Ehn et al., 2014; Bianchi et al., 2019). In the autoxidation process, intramolecular H-shift reactions can convert an $RO_2$ into a carbon-centered radical with a hydroperoxide functionality. A new $RO_2$ can then be formed through the subsequent addition

of oxygen. In some precursor molecules, this process can repeat multiple times, and the rapid addition of oxygen in this process can eventually result in very high oxygen contents of the produced $RO_2$ (Ehn et al., 2014).



The final formation of a closed-shell molecule from $RO_2$ takes place through different termination reactions. For instance, $RO_2$ can decompose unimolecularly, e.g., by OH loss to form a carbonyl product (Rissanen et al., 2014). In addition, a number of bimolecular reactions can efficiently terminate $RO_2$ and form a variety of organic species. The main terminating reaction

partners in the atmosphere are NO, $HO_2$, and other peroxyl radicals (Vereecken and Francisco, 2012). In the specific reaction of an $RO_2$ with another $R'O_2$, ROOR' dimers can be formed (Berndt et al., 2018; Finlayson-Pitts and Pitts Jr, 1999).

Kaurene has an exocyclic double bond, as shown in Fig. 2, and can react with both $O_3$ and OH. In reaction with $O_3$, two separate molecules will form following the scission at the initial double bond. Out of the resulting $C_1$ and $C_{19}$ species, one will typically form a "primary" $RO_2$ and the other a closed shell carbonyl. The initial steps can be considered equivalent to the

ozonolysis of the monoterpene β-pinene, which also has a lone double bond attached to a ring structure (Zhang and Zhang, 2005). If the carbonyl formed on the $C_1$ fragment, we expected to produce a $C_{19}H_{29}O_3$ peroxyl radical. In addition to the oxidation by $O_3$, also OH is of importance, as it is formed during the experiments as a by-product of kaurene-$O_3$ reactions. Kaurene oxidation by OH initiate via either addition of OH to the double bond, or abstraction of an H atom. Through OH attachment, the primary $RO_2$ will be $C_{20}H_{33}O_3$, and H abstraction by OH will result in $C_{20}H_{31}O_2$ as the primary $RO_2$. These

three radicals, initiated by ozonolysis and OH oxidation, are expected to be the starting points for the majority of observed large oxidation products, though no mechanistic studies exist on the radical reactions in kaurene oxidation. In this study, the potential of the radicals to undergo autoxidation and achieve higher oxygen contents was of particular interest. Oxidation products with five or more oxygen atoms are classified as HOM in this work.

### 2.4.2 HOM quantification

Based on earlier observations, HOM formed from kaurene oxidation are expected to be efficiently detected by the nitrate CI-APi-TOF (Bianchi et al., 2019). However, the approaches for valid and suitable calibrations still remain limited for these highly oxygenated and reactive molecules, severely hampering their reliable quantification. For estimation of HOM concentrations in this study, we adopted a similar approach as previously reported (Jokinen et al., 2012; Ehn et al., 2014; Jokinen et al., 2014) to convert the measured HOM ion signals to HOM concentrations using the following equation:

$$[HOM] = C \times \frac{\sum_i HOM_i \cdot NO_3^-}{NO_3^- + HNO_3 NO_3^- + (HNO_3)_2 NO_3^-} \tag{4}$$

Here, [HOM] is the estimated concentration of HOM in the chamber, C is a calibration factor determining the sensitivity, and $HOM_i \cdot NO_3^-$ is the signal intensity of a molecule classified as HOM and measured by the nitrate CI-APi-TOF as a cluster with $NO_3^-$. This approach implicitly assumes an identical sensitivity to all HOM. The ionization of the majority of the HOM is expected to proceed at the collision limit (Bianchi et al., 2019; Ehn et al., 2014; Hyttinen et al., 2015), but factors like mass-

dependent transmission (Ehn et al., 2011; Heinritzi et al., 2016) of the instrument may cause additional uncertainties. Still, the main challenge is to determine the calibration factor C, accounting for both charging efficiency and HOM losses in the sampling lines. In this study, we chose $C=1\times10^{10}$ $cm^{-3}$ based on typical values reported earlier (Jokinen et al., 2012; Jokinen et al., 2014; Ehn et al., 2014). This approach comes with significant uncertainties, estimated to be at least a factor 2 (+100%/-





50%). A strongly mass-dependent ion transmission would be the most likely reason for a clearly larger deviation. Nevertheless,

given the large uncertainty also in the kaurene concentration, we only use the calibration factors to approximate the HOM concentrations and yields.

## 2.5 Molar yield calculation

The change rate of HOM concentration in our chamber was determined by the production and loss rates according to Eq. (5):

$$\frac{d[HOM]}{dt} = \text{Production}_{HOM} - \text{Loss}_{HOM} \tag{5}$$

HOM can be formed both from reactions with $O_3$ and OH. Most of the time, the kaurene concentration was very low in the chamber and may not have been the major sink for the formed OH. Instead, OH was likely lost to reactions with $O_3$ and potential contaminant molecules in the chamber. In addition, it is also hard to estimate the relative importance of the two oxidants because neither the $O_3$ nor OH reaction rate coefficients with kaurene are well known. Thus, for simplicity, the production of HOM in the equation above will be written as purely the ozonolysis reaction, i.e., $k_1\gamma[\text{kaurene}][O_3]$, where $k_1$ is

the kaurene-$O_3$ reaction rate coefficient, and $\gamma$ is the HOM molar yield from the kaurene-$O_3$ reaction. Note that $\gamma$ will be overestimated if the kaurene-OH reactions contribute substantially to the measured HOM. The kaurene-$O_3$ reaction rate coefficient $k_1$ was assumed to be $5\times10^{-16}\text{cm}^3\text{molec}^{-1}\text{s}^{-1}$ as shown in Sect. 2.3.

The loss of HOM mainly consists of three parts: chemical reactions, flush-out from the chamber, and the condensation onto the walls and potential aerosol particles, and they can be written as $k_{loss}[HOM]$, where $k_{loss}$ is the total loss coefficient of HOM.

Under steady state, the concentration of HOM is constant, and we can write it as:

$$\frac{d[HOM]}{dt} = k_1\gamma[\text{kaurene}][O_3] - k_{loss}[HOM] = 0 \tag{6}$$

And thus,

$$\gamma = \frac{k_{loss}[HOM]}{k_1[\text{kaurene}][O_3]} \tag{7}$$

All data used to estimate HOM molar yield were selected from the periods when there were no seed particles in the chamber.

According to an earlier study in the COALA chamber by Peräkylä et al. (2020), the lifetime of HOM in the gas phase under a typical situation without seed particles could be assumed to be 200 s, primarily controlled by losses to chamber walls. Therefore, $0.005 \text{ s}^{-1}$ was used for $k_{loss}$ here. The flush-out rate is over an order of magnitude slower, and considering the low oxidant concentrations in our chamber, it is unlikely that the chemical loss of HOM would be faster than the wall loss rate.

## 3 Results

The ozonolysis of kaurene in the gas phase was investigated in the COALA chamber under different oxidation conditions. The $O_3$ concentrations ranged from ~70 ppb to ~110 ppb for different experiments, and the kaurene concentration was at most ~20 ppb, though most of the time well below 1 ppb (Fig. 3(a)). Seed particles were introduced to the chamber in some experiments





(the gray shades in Fig. 3), and the total particle mass reached its maximum value of more than 200 µg m$^{-3}$ on the January 31st experiment. In the following, the characteristics of oxidation products from kaurene ozonolysis, including chemical

identification, are first described (Sect. 3.1), followed by HOM molar yield estimates (Sect. 3.2). Sect. 3.3 discusses the volatilities of the oxidation products, and Sect. 3.4 depicts fragmentation patterns of kaurene and the formation of $NH_4^+$ clusters in the Vocus.

### 3.1 HOM and other oxidation products of kaurene ozonolysis

We observed the oxidation products of kaurene ozonolysis in the gas phase using Vocus and nitrate CI-APi-TOF. Product

distributions are shown in Fig. 4 including both less oxidized species and HOM, and the temporal behavior of some important oxidation products are shown in Fig. 3. When discussing detected compounds (M) in the following sections, they were either detected after protonation (MH$^+$) in the Vocus, or as clusters with nitrate ions (M·NO$_3^-$) in the CI-APi-TOF. In other words, the identified species are presented after omitting the reagent ions, but if the specific ions need to be addressed, they will contain the charging ion as well as the label for the charge ($^+$ or $^-$). When masses are discussed, it will refer to the mass-to-

charge ratio in the spectra where the compound was identified, including the corresponding reagent ion.

We first compared the spectra of the reactant-free chamber (no kaurene or O$_3$ injection) with that after injecting ~5 ppb kaurene. As shown in Fig. 4(a), a predominant peak of kaurene $C_{20}H_{32}$ along with a series of hydrocarbon ions was detected by the Vocus. These hydrocarbons with carbon numbers from 10 to 16 were observed between 130 Th and 230 Th. The time series of those hydrocarbons showed high similarities to the kaurene time series with very high correlation coefficients (R$^2$>0.95) at

different FIMR pressures, indicating that they most likely are fragments of kaurene formed in the Vocus. More details about the fragments of kaurene in the Vocus will be discussed in Sect. 3.4.

After introducing O$_3$ (~70 ppb), several less oxidized kaurene oxidation products ($C_{18-20}H_{28-34}O_{1-4}$, complete peak list in Table S3.) were observed between 270 Th and 330 Th in the Vocus spectra (Fig. 4(b)). The highest concentrations were observed from compounds identified as $C_{19}H_{30}O$, $C_{19}H_{32}O_2$, and $C_{20}H_{32}O_3$, of which the time series is shown in Fig. 3. Compounds with

identical elemental compositions have been observed in the headspace samples of heated pine needles and spruce twigs (Helin et al., 2020), but to the best of our knowledge, these compounds have not been reported in the ambient air. The ions containing N-atoms are believed to be adducts with $NH_4^+$, as will be discussed in Sect. 3.4.

In the nitrate CI-APi-TOF, various HOM monomers ($C_{18-20}H_{28-34}O_{5-13}$) and dimers ($C_{37-40}H_{58-64}O_{6-10}$) were detected in the mass to charge ratio (m/z) ranges 400-550 Th and 650-750 Th, respectively (Fig. 4(c) and (d)). Several radicals with $C_{19}$ and $C_{20}$

carbon skeletons were observed as well (Table S4). Overall, the spectra contain a wide range of molecules with different compositions, and some speculation about possible formation pathways is given below. However, we stress that our data is not optimal for deducing detailed oxidation mechanisms, and in some cases, several pathways may lead to the same product, while for some other products, no viable formation pathways were identified. Among the closed-shell HOM monomers, $C_{19}H_{28}O_6$, $C_{19}H_{28}O_8$, $C_{19}H_{30}O_7$, $C_{20}H_{30}O_5$, $C_{20}H_{30}O_7$, and $C_{20}H_{32}O_7$ were the most prominent signals. $C_{19}H_{28}O_{6,8}$ and $C_{19}H_{30}O_7$

could be explained by the primary ozonolysis RO$_2$ ($C_{19}H_{29}O_3$) undergoing autoxidation and then terminating either





unimolecularly (for the former) or via reaction with $HO_2$ (for the latter). Corresponding highly oxidized $RO_2$ were detected ($C_{19}H_{29}O_{7,9}$, Table S4). However, $HO_2$ is expected to be low in our chamber, so other formation pathways (e.g., $RO_2 + RO_2$ reactions) may be involved in forming $C_{19}H_{30}O_x$ species. Among the OH-initiated HOM, we detected $C_{20}H_{30}O_{5-8}$ as the dominant group, and related $RO_2$ were also identified ($C_{20}H_{31}O_{7-8}$, Table S4). $C_{20}H_{32}O_{6-9}$ HOM presumably initiated from OH-addition reactions were observed with decent signal intensities, with $C_{20}H_{32}O_7$ as the most abundant. However, signals of the corresponding $C_{20}H_{33}O_x$ radicals were very low, and only noisy signals were detected even at high kaurene concentrations.

The signal intensities of HOM dimers were about one to two orders of magnitude lower than those of monomers (Fig. 4(d)). The highest peaks in the dimer range were observed at 674 Th, 688 Th, 690 Th, 706 Th, and 722 Th corresponding to the formulae as $C_{38}H_{60}O_6$, $C_{39}H_{62}O_6$, $C_{38}H_{60}O_7$, $C_{38}H_{60}O_8$, and $C_{39}H_{64}O_8$, respectively. $C_{38}H_{60}O_{6,7,8}$ dimers are most likely produced from the cross-reactions of $C_{19}H_{29}O_x$ and $C_{19}H_{31}O_y$. Similarly, $C_{20}H_{33}O_{2-8}$ radicals could take part in different bimolecular reactions with $C_{19}H_{29}O_{2-6}$ and/or $C_{19}H_{31}O_{2-8}$ forming $C_{39}H_{62}O_6$ and $C_{39}H_{64}O_8$ dimers, respectively. While we do not have suggestions on the formation mechanisms of $C_{19}H_{31}O_x$ radicals, both these dimer signals and the monomeric $C_{19}H_{30}O_x$ species would most easily be explained by the involvement of these radicals. Another unanticipated finding was that we did not detect any dimers with oxygen numbers above 10, and the most abundant dimer signals were identified as compounds with 6–8 oxygen atoms. This is similar to the detected monomers, which is in contrast to earlier findings of monoterpenes and sesquiterpene (Quéléver et al., 2019; Jokinen et al., 2016; Li et al., 2021; Barreira et al., 2021; Kirkby et al., 2016; Rissanen et al., 2014), where dimers typically contained clearly more oxygen than the monomers. We can only speculate on the reasons for this, and a possible explanation is that the most oxygenated $RO_2$ have very short lifetimes and thus terminate through unimolecular channels before colliding with another $RO_2$. It is also possible that the nitrate CI-APi-TOF might be more sensitive towards less oxygenated dimers if there are 40 C-atoms rather than 20 in their skeleton, if a larger molecule increases the likelihood of forming a stable cluster with $NO_3^-$. Mass-dependent transmission may also play a role, as the detected dimers were around 700 Th and above, where sensitivity is most likely decreasing (Heinritzi et al., 2016).

Interestingly, a group of HOM with 21-29 carbon atoms was observed in the m/z range from 500 Th to 560 Th (Fig. 4(c)), and they exhibited similar temporal behaviors to those of other HOM (Fig. 3). The dominant peaks were identified as $C_{24,26,27,28}H_{36,38,40,42}O_{6-8}$ species. We conclude that the most probable explanation for these HOM is that they were the dimers formed via the reactions of kaurene-derived $RO_2$ with smaller $RO_2$ derived from various $C_{2-9}$ organic contaminants in the chamber. Both Vocus and nitrate CI-APi-TOF observed such contaminants during the experiments (Fig. S2), the likely source being fairly high-loading monoterpene experiments performed in the chamber prior to the kaurene experiments. Low-volatile gases and particles deposited on the chamber walls can then lead to compounds slowly off-gassing from the walls. In reactions with OH, these $C_{2-9}$ precursors can then form $RO_2$ that terminate by reacting with the $C_{19-20}$ radicals, ending up forming the observed $C_{21}$-$C_{29}$ HOM dimers. As most of our experiments were done with very low kaurene concentrations (<1 ppb), the contaminants could amount to an equally high, or even higher, OH sink than the kaurene itself. By comparing the most abundant observed contaminants and the $C_{24}$-$C_{28}$ dimers, we also tried to assess which $RO_2$ from kaurene were the most important. In most cases, $C_{19}H_{29}O_3$ and $C_{19}H_{31}O_2$ seemed to be the most plausible radicals to explain the most abundant



observed dimers. These were also thought to be the main contributors to $C_{38}$-$C_{39}$ dimers of kaurene oxidation. Unfortunately, due to their low oxygen content, we cannot directly detect these radicals with nitrate CI-APi-TOF.

During most of the experiments, all HOM detected by nitrate CI-APi-TOF exhibited similar temporal behaviors as the kaurene time series (Fig. 3), as expected due to the low variability in $O_3$ concentrations. The increase of products in the nitrate CI-APi-TOF reacted rapidly to changes in the injected kaurene, while the less oxidized products detected by the Vocus showed a

slower response to kaurene changes, especially when kaurene was removed.

## 3.2 HOM molar yield estimation

We calculated the molar yield for HOM from kaurene ozonolysis (without OH-scavenger) as described in Sect. 2.5. Figure 5 shows the measured HOM concentration as a function of the kaurene ozonolysis rate ($k_1 * [kaurene] * [O_3]$). In the kaurene ozonolysis experiments at low oxidation rates (Fig. 5(b)), the total HOM concentrations had a near-linear dependence on the

amount of kaurene reacting with $O_3$. We can estimate a molar HOM yield of ~2% in this range, though keeping in mind the large uncertainties in the quantification of kaurene and HOM. At higher oxidation rates (Fig. 5(a)), the increase of total HOM concentration exhibited a logarithmic dependence. The slower growth of HOM concentration at higher oxidation rates might be explained by increasing sinks in the chamber, e.g., due to particle formation. The $RO_2$ lifetimes will also decrease, which may hamper the autoxidation when bimolecular reactions become faster. When the entire data range was included, the molar

HOM yield ranged between 0.1%−3%.

Although large uncertainties are included in the derivation of our HOM yield from kaurene ozonolysis, yields of a few percent have been reported for a variety of other BVOCs earlier (Table 1). More studies are needed to better quantify the HOM formation in this system and get further chemical insights on the detailed oxidation pathways. Although the global emission budgets of diterpenes remain unquantified, combining the limited emission rates reported previously (Matsunaga et al., 2012;

Yáñez-Serrano et al., 2018) with the HOM yield estimated here, it is possible that HOM formation from diterpenes, and their influence on SOA formation, are of larger importance than previously thought.

## 3.3 Volatilities of the oxidation products

To deduce the ability of the oxidation products from kaurene ozonolysis to contribute to aerosol formation, we investigated their behavior as a function of the available condensation sink (CS) in the chamber. In this experiment, kaurene, $O_3$, and seed

particles were injected into the chamber until it reached a steady state (referred to as S1 in Fig. 3(c)). The injection of seed particles was then stopped, causing a decrease in the CS and a corresponding increase in the concentrations of several compounds, ultimately reaching a new steady-state condition (referred to as S2 in Fig. 3(c)). For each observed compound, we then calculated the concentration ratio at S1 and S2, henceforth termed the fraction remaining, FR (Fig. 6).

From Fig. 6(a), we found a clear transition of volatilities when the molecular mass increased, in line with previous findings

for α-pinene and cyclohexene systems (Peräkylä et al., 2020; Räty et al., 2021). Interestingly, all species detected by the Vocus had an FR around 1, i.e., not affected by the CS, suggesting that the Vocus primarily detects VOCs and IVOCs (intermediate-





volatility organic compounds). Kaurene (and all its fragments) were in this range, as expected. Signals with FR much above unity may either be semi-volatile compounds that never reached a steady state during the experiments, or molecules that were preferentially formed in the particle phase before evaporating into the gas phase. A sharp drop in the FR, from ~1 to ~0.1, was

found in the mass range 300–350 Th, and the oxidation products with more oxygen tended to have lower values. This trend is consistent with that in the mass range above 350 Th (Fig. 6(b)), where most HOM were found. This indicates that these species condensed efficiently, and were mainly LVOCs (low-volatility organic compounds) or ELVOCs (extremely low-volatility organic compounds). Similar results were reported for HOM formed from α-pinene ozonolysis by Peräkylä et al. (2020). The relatively large spread of the FR above 350 Th may indicate that the seed particles also impacted the formation pathways, in

addition to providing a larger CS. For example, some longer-lived $RO_2$ might also have been lost to the particles, as discussed by Peräkylä et al. (2020).

In the mass range 200–300 Th, the FR of nitrate CI-APi-TOF species were also around one even though some have quite high oxygen content (Fig. S3(a)). These compounds were primarily contaminants from the chamber walls, typically with less than ten carbon atoms (Fig. S3(c)). While the high O-atom content might suggest lower volatility, the FR around unity suggests

that the CS was not a noticeable sink for these species. On the one hand, this is credible since their existence in the chamber during these experiments means that they evaporate efficiently from the chamber walls. On the other hand, it is also possible that there is large storage on the walls, and they have reached a balance between the gas-phase and wall concentrations. When these species condensed onto the seed particles, more came out from the chamber walls, keeping the measured gas-phase concentrations stable. Either way, although not of particular importance for this study, the mere existence of these compounds

in the continuously flushed chamber strongly suggests that they are SVOCs (semi-volatile organic compounds).

In summary, these results show that the oxidation products of kaurene ozonolysis with ~5 or more oxygen atoms were mainly LVOCs or ELVOCs, and will thus efficiently condense to form organic aerosols. In line with earlier studies (Räty et al., 2021), we also found that the oxygen content is the most important parameter affecting the volatility. Despite the longer carbon chain, and thus higher mass, of the diterpenes, compared to, e.g., monoterpenes or cyclohexene, the transition from condensing to

non-condensing was not dramatically different. For those compounds ($C_6$ and $C_{10}$), similar experiments indicated that ~7 O-atoms were needed, whereas, for the $C_{20}$ kaurene, the number was around 5.

## 3.4 Fragmentation and $NH_4^+$ clusters in Vocus PTR-TOF

Fragmentation is a ubiquitous process when measuring VOCs with PTR instruments, complicating the identification and interpretation of the mass spectra. Previous studies have shown that the settings of PTR-MS (in essence E/N), the types of

VOCs, and environmental conditions (e.g., RH) can all affect the fragmentation process and change the fragment distributions (Tani et al., 2003; Demarcke et al., 2010; Kim et al., 2010; Rimetz-Planchon et al., 2011; Gueneron et al., 2015). Within the standard E/N settings (80 Td–140 Td) of PTR instruments, a decrease of E/N may decrease the absolute sensitivity, although the fraction of signal detected at the parent ion may become larger due to decreased fragmentation (Kim et al., 2009; Demarcke et al., 2010). Various terpenes, including monoterpenes and sesquiterpenes, have been observed to undergo different degrees





of fragmentation within the PTR instruments (Tani et al., 2003; Kim et al., 2009; Rimetz-Planchon et al., 2011; Misztal et al., 2012). However, potential fragmentation patterns of diterpenes have not been reported before this study. We find several fragments of kaurene in the Vocus, and exploiting the decreases in FIMR pressures, we were able to investigate the fragmentation process of kaurene as a function of this pressure as well.

As can be seen in Fig. 4(a), a total of seven main kaurene fragments with carbon numbers exceeding ten were observed in the

Vocus spectra. Also, some peaks smaller than 130 Th in the spectra were seen. However, as the difference in their signals before and after kaurene injections are very small, we focus on the seven main fragments in this section. All fragments were detected as protonated ions ($MH^+$). We classified these fragments into two groups (Group 1: $C_{10}H_{15}^+$, $C_{11}H_{17}^+$, $C_{12}H_{19}^+$, $C_{13}H_{21}^+$, $C_{14}H_{23}$;$^+$; Group 2: $C_{15}H_{23}^+$, $C_{16}H_{25}^+$), which each included fragments separated by a $CH_2$ unit (14 Th) based on a previous classification method reported by Sovová et al. (2011). The fragments peaks were dominated by Group 1, and the ratio of

Group 1 signals to Group 2 signals was around 6 for all the experiments. When FIMR pressure was ~1.4 mbar (original setting in this study), m/z 191 $C_{13}H_{21}^+$ was the highest signal (Fig. 7(h)), and the ratios of $C_{13}H_{21}^+$ signal to $C_{11}H_{17}^+$, $C_{12}H_{19}^+$, and $C_{14}H_{23}^+$ were about 1.6, 1.3, and 3, respectively. m/z 217 $C_{16}H_{25}^+$ was the dominant signal in Group 2, and it was 2-fold higher than that of $C_{15}H_{23}^+$. All these fragments of kaurene were much larger than those of monoterpenes (Maleknia et al., 2007; Tani et al., 2003; Misztal et al., 2012) because the precursor kaurene is twice as larger as monoterpenes ($C_{10}H_{16}$) in terms of

molecular mass. However, the fragment m/z 149 $C_{11}H_{17}^+$ was observed as the most abundant fragment of many sesquiterpenes in previous studies (Kim et al., 2010; Kim et al., 2009; Demarcke et al., 2010). Table 2 summarizes fragmentation patterns for kaurene (in this study) and sesquiterpenes from previous studies (Kim et al., 2009). Although only one fragment ($C_{11}H_{17}^+$) was identical between kaurene and the sesquiterpene, we found that lost neutral fragments were surprisingly similar between the di- and sesquiterpenes.

Figure 7 (a)-(g) show the relationship between kaurene ($C_{20}H_{33}^+$) signals and its fragments under different FIMR pressures. From these figures we can see that the signals of fragments at FIMR pressure of 0.9 mbar increased more than one order of magnitude relative to $C_{20}H_{33}^+$ compared to those at 1.4 mbar for $C_{11}H_{17}^+$, $C_{12}H_{19}^+$, and $C_{15}H_{23}^+$. The increase of $C_{13}H_{21}^+$ and $C_{14}H_{23}^+$ were smaller, but they still doubled when the pressure dropped from 1.4 mbar to 0.9 mbar. In addition, the ratio of the sum of all fragments to $C_{20}H_{33}^+$ increased from ~0.4 to ~3 during the experiments. Thus, it is clear that the fragmentation

increases with decreasing FIMR pressures, as can be expected since the electric field strength stayed constant, thus leading to a net increase in E/N.

Changes in FIMR pressure also impacted the distribution of the fragments (Fig. 7(h)). When Vocus was run at the FIMR pressure of 1.4 mbar, $C_{13}H_{21}^+$ was the most abundant fragment signal. However, when the pressures decreased, the fractions of $C_{13}H_{21}^+$ started to decrease, and at the end of the experiments (FIMR pressure = 0.9 mbar), $C_{12}H_{19}^+$ and $C_{11}H_{17}^+$ had become

the dominant fragments. The overall distribution of fragments for Group 1 clearly transitioned towards smaller fragments, with all the largest ones decreasing while the smaller ones increased. Though fewer ions, the same trend was also visible in Group 2.





In addition to the protonated molecular ions and fragments typically observed in PTR instruments, some compounds were also found to be charged by ammonium, $NH_4^+$, which can be intrinsically formed in the ion source (Norman et al., 2007; Müller et

al., 2020), or be introduced as impurities either in the water or sample air. Although typically not a major signal, in this study, we observed the clusters of kaurene and two oxidation products with $NH_4^+$ at surprisingly high amounts (Fig. 4 (a)). As shown in Fig. 8, the temporal behavior of the $NH_4^+$ adducts was in good agreement with those of the corresponding protonated ions ($R^2 > 0.9$). The ratio between the $NH_4^+$ cluster signal and the protonated signal was less than 2% for $C_{20}H_{32}$, whereas the ratios could be more than 50% (ranging from 15% to 70%) for the two oxidation products, $C_{19}H_{30}O$ and $C_{19}H_{30}O_2$. These ratios

decreased over time, as the FIMR pressure decreased, and subsequently the collision energies, and thus the fragmentation, increased.

We cannot say if $NH_4^+$ production in our Vocus during this study was higher than typical, as the signal is buried under the large water signals. However, it is possible that the large size of the diterpene and its oxidation products may help to stabilize the clusters formed with $NH_4^+$, thus increasing the survival probability compared to $NH_4^+$ adducts with smaller molecules. In

addition, if protonation of these large molecules would lead to considerable fragmentation, $NH_4^+$ adducts, as a softer ionization method (Chen and Her, 1993; Harrison, 2018), may again increase the probability of the adduct compared to the protonated molecule. Thus, it is possible that an instrument with $NH_4^+$ ionization could be better suited for the detection of diterpenes and their oxidation products, but further studies are needed to validate this hypothesis.

**4 Conclusions**

This study presents the first characterization of gas-phase oxidation products from diterpene ozonolysis. We studied the reaction of the diterpene kaurene with $O_3$ in a simulation chamber, monitoring the products using a Vocus PTR-TOF and a nitrate ion-based CI-APi-TOF. Less oxidized kaurene oxidation products, $C_{18-20}H_{28-34}O_{1-4}$, were observed with the Vocus, while $C_{18-20}$ HOM monomers with carbon skeleton and $C_{37-40}$ dimers were detected with the nitrate CI-APi-TOF. In addition, several corresponding $RO_2$ were detected during the experiments. The most abundant monomers were $C_{19}H_{28}O_8$ and $C_{20}H_{30}O_5$,

and the highest signals of dimers were identified as $C_{38}H_{60}O_6$ and $C_{39}H_{62}O_6$. A group of HOM with carbon numbers in the range $C_{21}$-$C_{29}$ was surprisingly observed in this study, and their temporal behaviors were similar to those of other oxidation products. These $C_{21}$-$C_{29}$ HOM were assumed to be accretion products formed via the reactions of kaurene-derived $RO_2$ with some smaller $RO_2$ formed from OH oxidation of evaporated contaminants from the chamber walls. The HOM molar yield from kaurene ozonolysis without using an OH scavenger was estimated to be ~2%, comparable to those reported for other terpenes,

though the exact value comes with a large uncertainty due to the uncertain quantification of both the HOM and the precursor. The volatilities of the oxidation products formed in kaurene ozonolysis were also investigated by adding seed aerosol as an additional CS. None of the species detected by the Vocus seem to be affected by the increasing CS, indicating they were VOCs and IVOCs. However, most HOM, with five or more oxygen atoms, detected with the nitrate CI-APi-TOF were assumed to be LVOCs or ELVOCs because they efficiently condensed onto the seed particles.



For detecting diterpenes in air, the exact behavior of the VOCs in a PTR instrument is of importance. Here, fragment distributions of kaurene in a Vocus PTR instrument were reported. We observed seven main kaurene fragments: $C_{10}H_{15}^+$, $C_{11}H_{17}^+$, $C_{12}H_{19}^+$, $C_{13}H_{21}^+$, $C_{14}H_{23}^+$, $C_{15}H_{23}^+$, and $C_{16}H_{25}^+$. The lost neutral fragments are consistent with those observed earlier for sesquiterpenes. The fragmentation process of kaurene was found to be a function of the Vocus FIMR pressure, with the fragment signals increasing rapidly and even surpassing that of the parent ion. In addition, the fragment distribution shifted

towards smaller fragments as the FIMR pressure decreased. Interestingly, we also observed that kaurene and two oxidation products were detected as clusters with $NH_4^+$, in addition to the typical protonated molecular ions in the Vocus. The ratio between the $NH_4^+$ cluster signal and the protonated signal was less than 2% for $C_{20}H_{32}$, but the ratios ranged from 15% to 70% for two oxidation products $C_{19}H_{30}O_{1,2}$. Future studies should evaluate the capability of different methods to detect diterpenes and the corresponding oxidation products, as it is possible that these larger molecules may be more efficiently detected as

clusters.

Based on the estimated HOM yield and low volatility of kaurene oxidation products, the influence of diterpenes on SOA formation may be more important than previously thought. However, more studies are required to determine the global emission budget of diterpenes. In addition, further work should try to understand the oxidation pathways and HOM contributions of other diterpenes in the laboratory and the ambient air, in order to produce a better picture of the role of

diterpenes in atmospheric SOA formation.

**Data availability.** Data is available upon request by contacting the corresponding authors.

**Author Contributions.** ME and AMYS initiated the study. ME and OG designed the experiments. YL, OG, HL, FG, YZ,

MM, and OP conducted the measurements and operated the chamber. YL performed the main data analysis of gas-phase and related model results supervised by ME. OG helped with kaurene-$O_3$ rate coefficient estimate performed by YL. FG analyzed the particle-phase data. APP and AL performed and analyzed the TORM measurement. YL plotted the main figures and wrote the original draft. All authors discussed the results and commented on the paper.

**Competing interests.** The authors declare that they have no conflict of interest.

**Acknowledgments.** The authors thank Toni Tykkä for the TD-GC-MS analysis of the Tenax tubes samples. This research has been supported by the European Research Council (grant no. 638703-COALA, and Synergy grant ERC-SyG-2013-610028, IMBALANCE-P), the Academy of Finland (grant nos. 317380, 320094, 307797 and 314099), the Spanish Government (grant

PID2019-110521GB-I00), Fundación Ramón Areces (grant ELEMENTAL-CLIMATE), and Catalan Government (grants SGR 2017-1005 and AGAUR-2020PANDE00117). Yuanyuan Luo was funded by China Scholarship Council.



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




**Table 1. The molar HOM yields from different atmospheric biogenic VOCs ozonolysis. All HOM were detected using nitrate CI-APi-TOF.**

| VOCs | HOM yield (%) | Source |
|---|---|---|
| cyclohexene | $4.0 \pm 2$ | Ehn et al. (2014) |
| | $4.5 \pm 3.8$ | Rissanen et al. (2014) |
| | 6.0 | Berndt et al. (2015) |
| α-pinene | $7.0 \pm 3$ | Ehn et al. (2014) |
| | 3.4 | Jokinen et al. (2015) |
| β-pinene | < 0.1 | Ehn et al. (2014) |
| | 0.12 | Jokinen et al. (2015) |
| limonene | $17.0 \pm 8.5$ | Ehn et al. (2014) |
| | 5.3 | Jokinen et al. (2015) |
| β-caryophyllene | $1.7 \pm 1.28$ | Jokinen et al. (2016) |
| | 0.5 | Richters et al. (2016) |
| α-humulene | 1.4 | Richters et al. (2016) |
| ent-kaurene | 2 (~0.5-10) * | this study |

* This range was roughly estimated based on the uncertainties in kaurene correction, rate coefficient of kaurene-$O_3$, and HOM quantification.

**Table 2. A summary of fragmentation patterns of diterpenes and sesquiterpenes. The E/N is ~120 Td for all studies listed here. The speculated loss represents the possible neutral molecules lost from the parent ion to form the detection fragments.**

| Speculated Loss | Kaurene ($C_{20}H_{32}$) * | | Sesquiterpene ($C_{15}H_{24}$) # | |
|---|---|---|---|---|
| | Fragment (m/z) | Detected fragment Ions | Fragment (m/z) | Detected fragment Ions |
| $C_{10}H_{18}$ | 135 | $C_{10}H_{15}^+$ | | |
| $C_9H_{16}$ | 149 | $C_{11}H_{17}^+$ | 81 | $C_6H_9^+$ |
| $C_8H_{14}$ | 163 | $C_{12}H_{19}^+$ | 95 | $C_7H_{11}^+$ |
| $C_7H_{12}$ | 177 | $C_{13}H_{21}^+$ | 109 | $C_8H_{13}^+$ |
| $C_6H_{10}$ | 191 | $C_{14}H_{23}^+$ | 123 | $C_9H_{15}^+$ |
| $C_5H_{10}$ | 203 | $C_{15}H_{23}^+$ | 135 | $C_{10}H_{15}^+$ |
| $C_4H_8$ | 217 | $C_{16}H_{25}^+$ | 149 | $C_{11}H_{17}^+$ |
| $C_5H_8$ | | | 137 | $C_{10}H_{17}^+$ |

*This study.

#Kim et al. (2009).





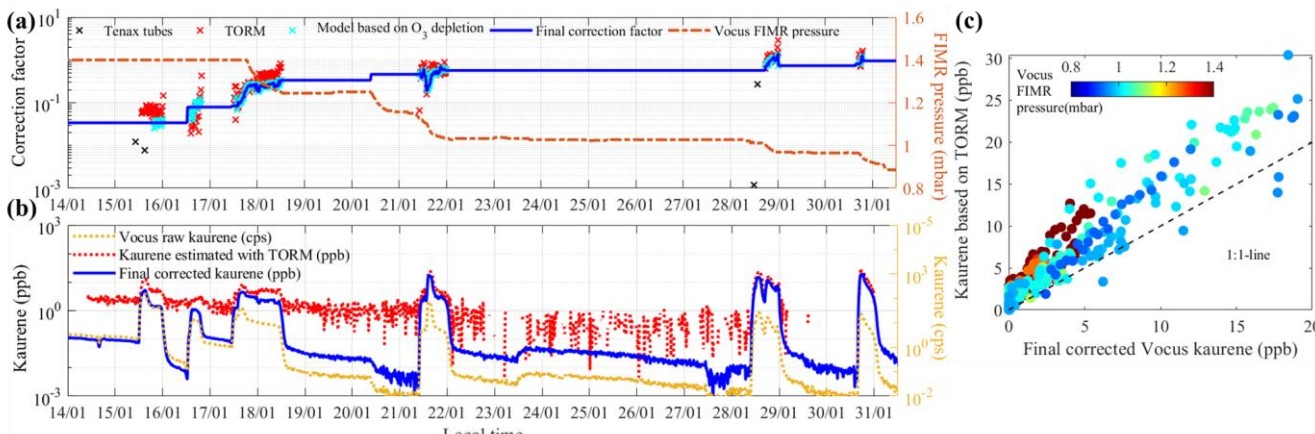

**Figure 1. Correction factors and time series of kaurene.** Panel (a) shows the correction factors (left axis) needed to scale the Vocus count rates to mixing ratios at varying FIMR pressures (right axis) using different methods. The black crosses represent the needed correction factors based on Tenax tube samples, the red crosses represent correction factors derived from the TORM system, and the blue crosses represent correction factors according to the $O_3$ depletion model (see Sect. 2.3 for details). The blue line shows the final correction factors we used in this study. Panel (b) displays the time series of the measured raw kaurene signal intensities (yellow), the kaurene concentration estimated by the $O_3$ reactivity system TORM (red), and the final kaurene concentration after corrections (blue). Panel (c) shows the relationship between the kaurene concentrations based on the TORM system and the final corrected Vocus kaurene concentrations.

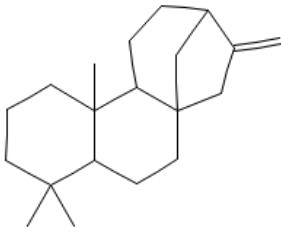

**Figure 2. The chemical structure of kaurene.**



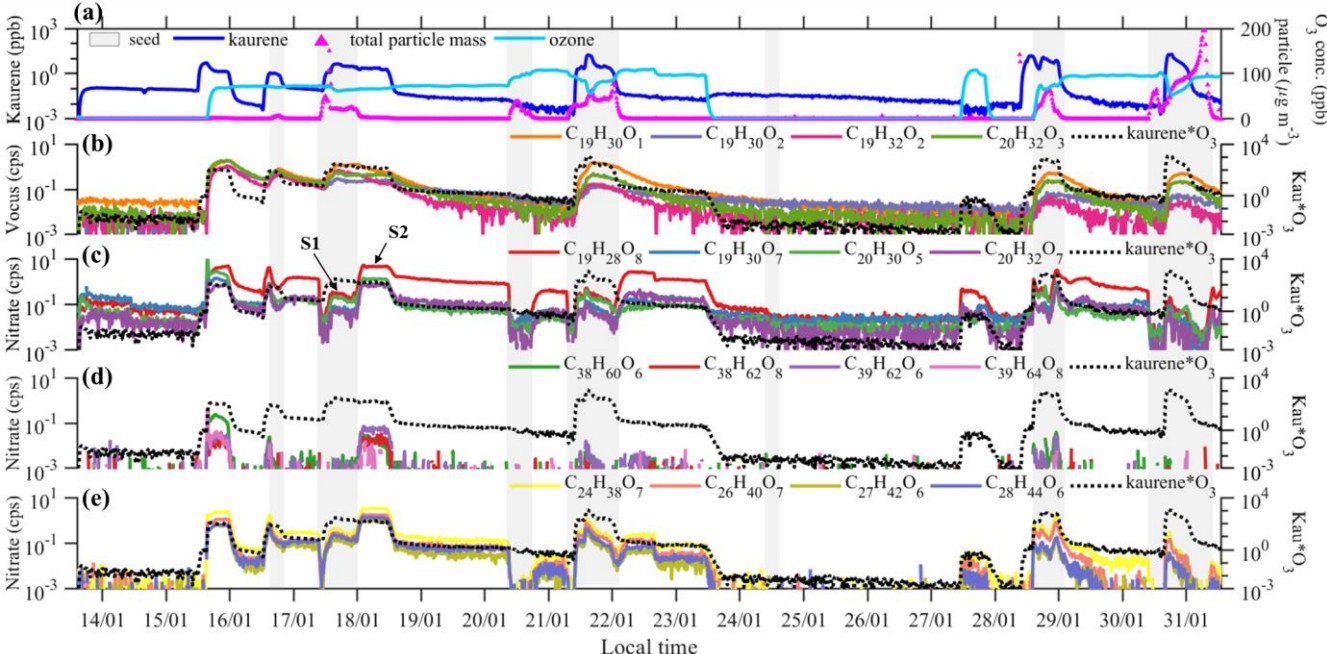

**Figure 3. Overview of the measurements from January 13th to 31st, 2020. Panel (a) shows O₃ and corrected kaurene concentrations and the total particle mass calculated from SMPS and AMS measurements. Some of the most abundant oxidation products measured with Vocus are shown in panel (b). HOM monomers and dimers of kaurene ozonolysis detected by nitrate CI-APi-TOF are shown in panels (c)–(e). The light gray shade indicates periods when there were seed particles in the chamber. The black dashed lines are included in each panel to depict the kaurene ozonolysis rate (Kau\*O₃ in the right y-axis representing kaurene concentration times O₃ concentration, in units ppb²). The 'S1' and 'S2' mark two steady-state periods used to determine the condensation behavior of oxidation products (Sect. 3.3).**





**Figure 4. Background-subtracted (diff) spectra of kaurene and kaurene oxidation products. The spectrum in panel (a) shows the difference between the Vocus spectra before and after kaurene injection; the spectrum in panel (b) displays the change of Vocus spectrum before and after O₃ was injected into the chamber with kaurene; Comparison of nitrate CI-APi-TOF spectra before and after O₃ injection into the kaurene-existing chamber is shown in panel (c) and (d). The two peaks labelled with * in panel (b) are the**






exact formula detected by Vocus, whereas the other species (M) in panels (a) and (b) are detected as MH⁺. All peaks labelled in
panels (c) and (d) are detected as a cluster with NO₃⁻.

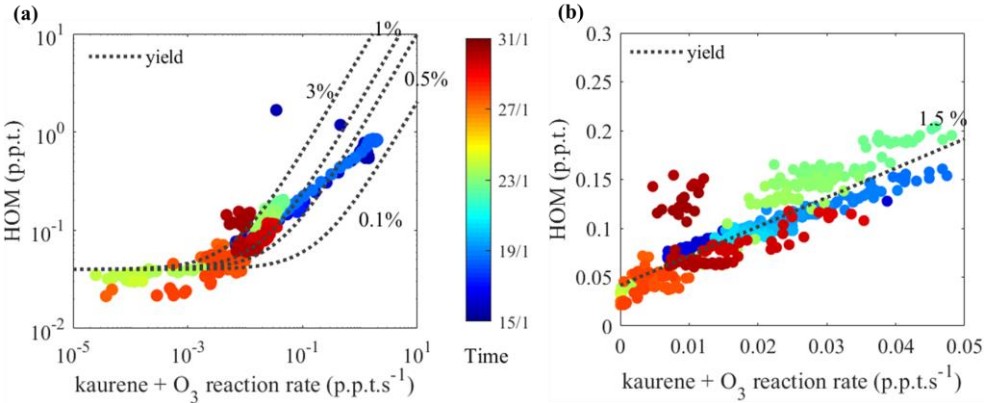

Figure 5. HOM yield estimation. HOM concentrations were plotted against the kaurene ozonolysis rate in panels (a) and (b). Markers
are colored by time, representing the changes of calibration factors applied to kaurene signals. Lines are added to the plots to
represent constant HOM molar yields from kaurene ozonolysis, accounting for an instrumental HOM background in the CI-APi-
TOF of 0.04 ppt. (a) All data points (plotted in logarithmic scale) from the periods without seed particles in the chamber. A zoomed-
in view of panel (a) with reaction rates ranging from 0 to 0.05 ppt s⁻¹ is shown in panel (b) (plotted in linear scale).

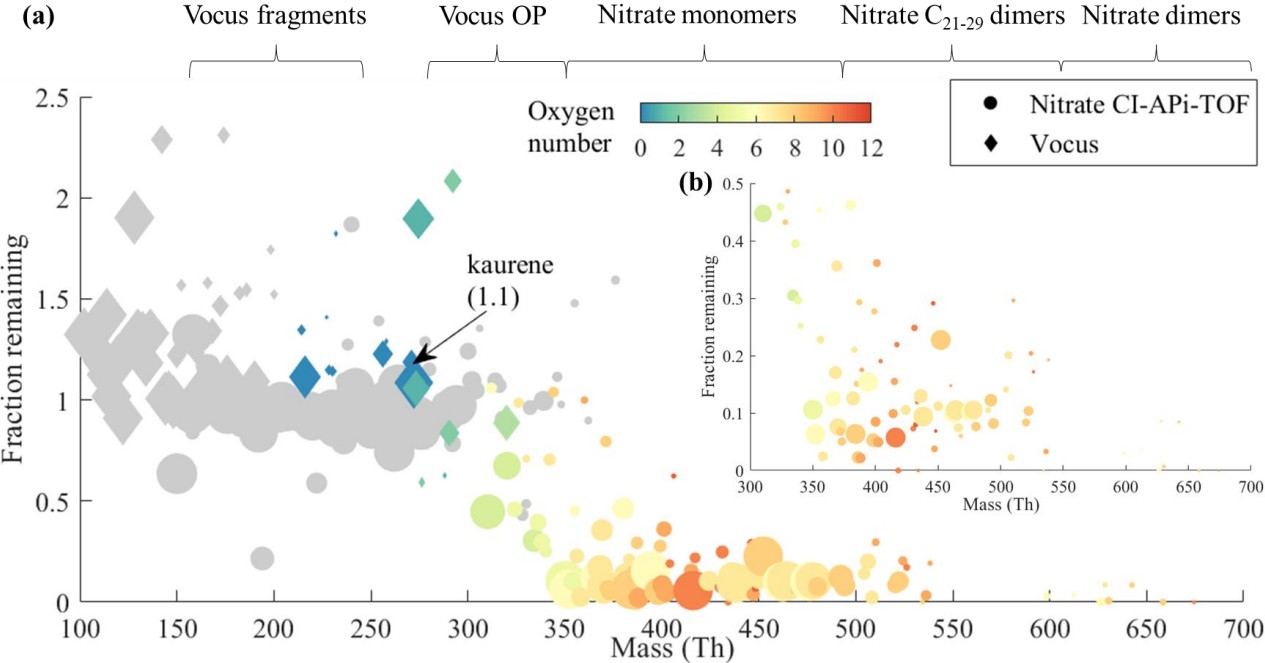

Figure 6. Seed injection behaviour. The "fraction remaining" corresponds to the ratio of the signal with seed particles to the signal
without seed. We emphasize that the molar mass indicated on the x-axis is without the reagent ions, unlike many previous studies
(Peräkylä et al., 2020; Räty et al., 2021). Species with less than 15 carbon numbers in the composition are colored with gray, while
the color scale indicates the oxygen content for species with 15 or more carbon atoms. The area of circles (nitrate CI-APi-TOF) and





diamonds (Vocus) is scaled linearly to the magnitude of each compound's signal when there were no seed particles in the chamber (different scaling for the two instruments). Compounds with the average signal intensities lower than three times the standard deviation are excluded from the plot. The text at the top shows roughly which compound groups are found in which parts of the mass range. The 'Vocus fragments' refers to the kaurene fragments, and 'Vocus OP' refers to $C_{18-20}$ oxidation products. 'Nitrate monomers', 'Nitrate $C_{21-29}$ dimers', and 'Nitrate dimers' represent the $C_{18-20}$ HOM monomers, $C_{21-29}$ HOM dimers, and $C_{37-40}$ HOM dimers measured with nitrate CI-APi-TOF, respectively. A zoomed-in view of the mass range above 300 Th is shown in panel (b).

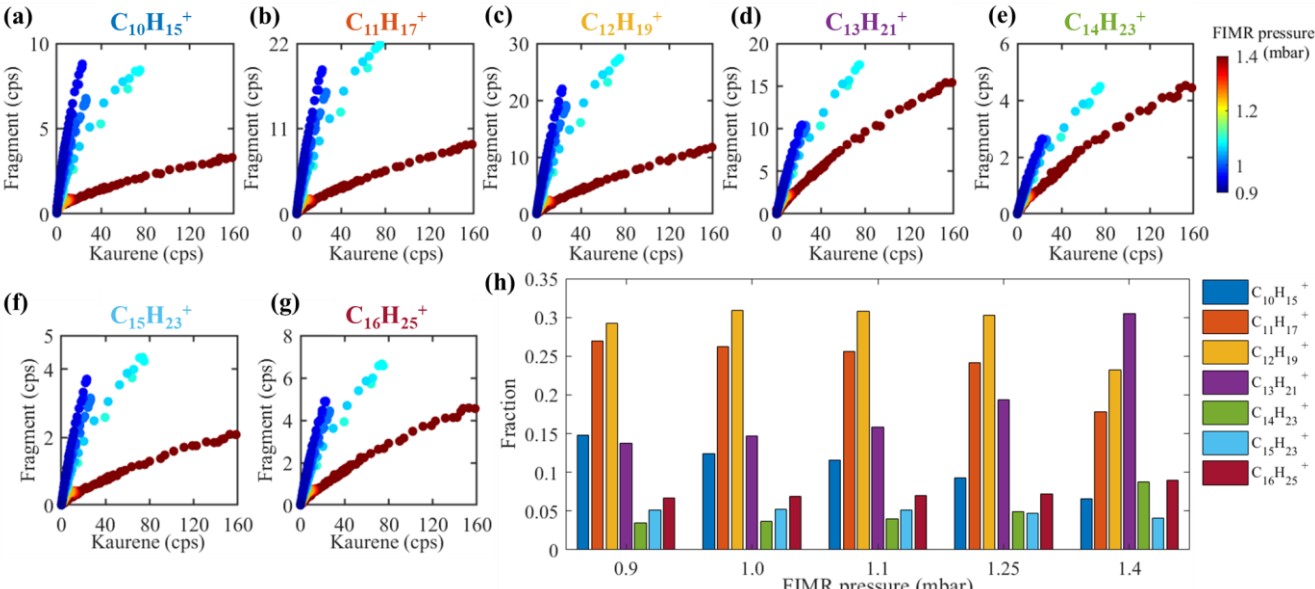

Figure 7. Fragments in Vocus. Panels (a)-(g) show the relationship between the signals of kaurene and its fragments in Vocus. Kaurene and its fragments are shown with raw signal intensities without corrections, colored with FIMR pressures ranging from 0.9 to 1.4 mbar. Panel (h) shows the fraction of the fragments as a function of FIMR pressures in Vocus. (fraction = signal of fragment / total signal of all fragments).





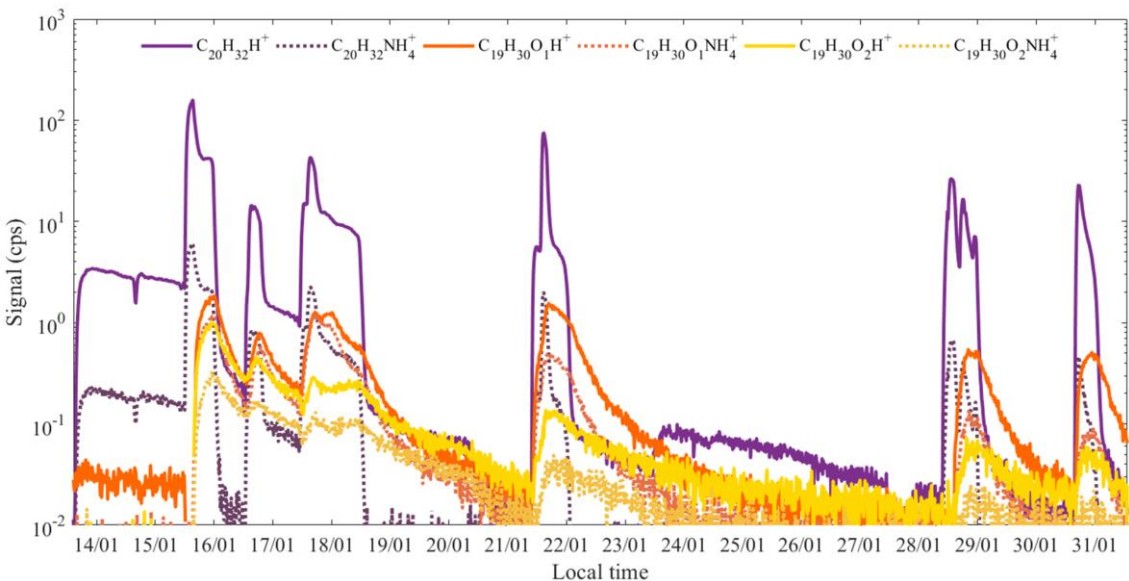

**Figure 8. Comparison of the measured time series of kaurene and two oxidation products detected as protonated or adducts with NH$_4^+$. Both the protonated and NH$_4^+$ cluster signals are raw data without FIMR pressure corrections.**