# Peer review of "Oxidation product characterization from ozonolysis of the diterpene ent-kaurene"

_Atmospheric Chemistry and Physics, 2021_

## Referee Comment (RC2)

MS No.: acp-2021-881

The authors describe experimental results from the ozonolysis of the diterpene kaurene ($C_{20}H_{32}$) carried out in a low-flow 2 m³ Teflon reactor for close to atmospheric conditions and RH < 1% with a residence time of the reaction mixture of about 1 hour. Kaurene was dosed by flushing a carrier gas through a heated vial filled with the hydrocarbon and introducing the flow into the reactor. Gas-phase analysis was carried out applying a Vocus PTR mass spec and a nitrate-based CIMS. A SMPS system and a AMS analyzed the formed particles. Resulting kaurene concentrations in the Teflon reactor were tried to determine using Tenax absorbers and by its $O_3$ reactivity with and assumed rate coefficient k($O_3$+kaurene). Generally, this manuscript describes a very challenging experimental work. A well-defined conversion of such a sticky compound in this reactor type is a really hard job.

This manuscript represents the first reported product study of a diterpene. Despite the big uncertainty of stated product concentrations and formation yields, the results are interesting and worth to be published in ACP. Some points should be considered before final acceptance is recommended.

1. Line 96: What is the reason using RH <1%?
2. Line 99: Is something known in the literature regarding vapor pressure and melting/boiling point of kaurene?
3. Line 155: Poor reliability of kaurene quantification is a weakness of this work. Did the authors think about a GC-FID technique applying the "effective carbon number" approach, which is especially suitable for pure hydrocarbons with a relatively high carbon number. It´s an old, but very robust, technique for measuring concentrations of hydrocarbons as successfully used for measuring sesquiterpenes in kinetic and product studies for atmospheric conditions. Should work for kaurene as well.
4. Line 183: I think the chosen rate coefficient k($O_3$+kaurene) = 5 x $10^{-15}$ cc/s cannot be estimated from modeling as described in the SM. There are 4 free parameters (kaurene feed, k($O_3$+kaurene), k(OH+kaurene) and OH yield) you can play with. There is not enough experimental information to fix them independent from each other.
   Is k($O_3$+kaurene) available via SAR?
   What about to measure k($O_3$+kaurene) using a relative rate technique? A signal, proportional to the kaurene concentration, is available from the PTR-ms.
   With the knowledge of a more reliable k($O_3$+kaurene) (and in the presence of an OH scavenger), a better estimate of the kaurene concentration should be possible based on the measured $O_3$ disappearance.
5. Line 230: A reaction scheme showing the expected first pathways would be helpful to better understand the argumentation given in this paragraph.
6. Line 252: Is the "calibration factor" c = $10^{10}$ molecules/cc not coming from sulfuric acid calibration? Is there an uncertainty only with a factor of 2? Was duty cycle correction applied? Additional uncertainty arising from the ion transmission of the big ions is mentioned. What is the expected total error in HOM concentrations?
7. Line 273: Also here, a (rough) estimate of the uncertainty in the HOM yield should be given.

---

## Author Comment (AC1)

**Response to Reviewers**

**RC1**

This manuscript describes chamber experiment results on the gas-phase reaction of ent-kaurene with ozone. Ent-kaurene represents one of the typical diterpenes that emit from biogenic sources. The authors have measured and quantified ent-kaurene using a PTR-CIMS and reported its fragmentation pattern. They have further characterized less oxygenated and highly oxygenated reaction products using a PTR-CIMS and a nitrate-CIMS, respectively. The results show that the total HOM yield from ent-kaurene ozonolysis is around 2 %, with the oxygen atom number for most of the monomers and dimers below 10. Further, the steady-state concentration of highly oxygenated products decreases as the mass loading of seed particles increases, while less oxygenated products are insensitive to the mass loading changes. The research topic of this paper is novel and the measurement techniques used for product characterization are state-of-the-art. Overall this is a very relevant study fitting perfectly into the scope of ACP.

However, the way the results are presented and discussed needs certain revisions in order to enhance the usefulness to the CIMS community and also be comprehensible for non-specialist readers. And the results would be more robust if the experiment quality could be improved, but I guess it is a bit unrealistic here. Nevertheless, here are my major remarks:

1. The authors have discussed some formation mechanisms when identifying the oxidation products. But I strongly encourage the authors to draw a reaction scheme for clarity. Otherwise, every reader would need to draw their own ones to make sense of the chemistry.

   Response: Thank you for pointing this out. We have added a reaction scheme to clarify the standard mechanisms leading to the primary $RO_2$ from kaurene-ozone and kaurene-OH reactions in Figure 3 (Page 24). We do not explicitly draw out any following reactions leading to closed-shell species, as our data cannot provide support for any particular steps among the many possibilities that exist.

Figure 3. Simplified mechanism of the formation routes of the primary $RO_2$ from kaurene-$O_3$ and kaurene-OH reactions. POZ: cyclic primary ozonide; CI: Criegee intermediates; SCI: stabilized Criegee intermediates; VHP: vinyl hydroperoxides.

2. Figure 3 is useful, but it is too busy to comprehend, and readers are actually not able to zoom in to figure out the details. It would make sense to just show a few time intervals instead of the entire period, and put this one in SI.

Response: We agree that it is not easy to see any specific details from this figure, but we nevertheless feel it is a valuable overview figure in order to provide a context for the results section. In particular the semi-volatile behavior of kaurene had a large impact on the experiments, and the e.g. for the HOM yield estimation, we mainly use periods without active addition of kaurene. As such, and given that we don't emphasize any specific time intervals in our study, we prefer keeping Figure 3 (Figure 1 in the revised manuscript) in the main text. However, we added a new figure (Figure S4) showing the time series of six important products from January 15th to 19th in the SI to make smaller details better visible to readers.

[Figure]

Figure S4. Measurements from January 15th to 19th, 2020. Panel (a) shows $O_3$ and corrected kaurene concentrations and the total particle mass calculated from SMPS and AMS measurements. Oxidation products measured with Vocus and nitrate CI-APi-TOF are shown in panel (b). The light gray shade indicates periods when there were seed particles in the chamber. The black dashed lines in panel (b) depicts the kaurene ozonolysis rate (Kaurene*$O_3$ in the right y-axis representing kaurene concentration times $O_3$ concentration, in units ppb$^2$). The 'S1' and 'S2' mark two steady-state periods used to determine the condensation behavior of oxidation products (Sect. 3.3).

3.  For Figure 4, the authors should probably either label the peaks with the reagent ion (e.g. M.NO3-) or subtract the mass of the reagent ion. The authors have written a disclaimer in the text, but it can still be a bit misleading.

    Response: This is a typical challenge when plotting mass spectra, as also in this case, adding "$NO_3$" to each label would make the plot extremely crowded, while only repeating the identical text 30+ times. On the other hand, removing the mass of the reagent ion makes the x-axis not correspond to the measured values, complicating comparisons with later experiments with the same instrument. We agree that there was a risk for misunderstanding, and now labelled the peaks detected by Vocus the actually detected ions in panels (a) and (b) (Figure 4). For the nitrate CI-APi-TOF panels (c) and (d) we suggest to still omit the reagent ion, but have added a textbox into each panel highlighting that all peaks are detected as clusters with $NO_3^-$. We hope that this is clear enough now, and since panels (a) and (b) are labelled to include the charges (+ sign), we expect that it will be even more evident that the neutral species labels in panels (c) and (d) are, in fact, missing the reagent ion.

[Figure]

Figure 4. Background-subtracted (diff) spectra of kaurene and kaurene oxidation products. The spectrum in panel (a) shows the difference between the Vocus spectra before and after kaurene injection; the spectrum in panel (b) displays the change of Vocus spectrum before and after $O_3$ was injected into the chamber with kaurene; Comparison of nitrate CI-APi-TOF spectra before and after $O_3$ injection into the kaurene-existing chamber is shown in panel (c) and (d). The compounds in panels (a) and (b) are labelled as what they were actually detected by Vocus, whereas all peaks labelled in panels (c) and (d) are detected as a cluster with $NO_3^-$.

4. The calibration factor of 1*10^10 cm^-3 for HOM quantification may not be that accurate, because the m/z of diterpene oxidation products are possibly 100-200 Th larger than those of monoterpene products. This is where the mass-dependent transmission does matter. The authors should at least show the transmission curve; and if the m/z of major products are sitting on the edge of the curve, a transmission coefficient needs to be included in the calculation of HOM concentration.

Response: The calibration factor will indeed vary due to mass-dependent transmission. However, unfortunately we do not have a transmission curve from which a transmission coefficient could be estimated. In the vast majority of studies, the calibration coefficient has been determined for $H_2SO_4$, and between 100 Th and the HOM ranges of MT and DT, the calibration coefficient can change considerably. At larger masses, like those between the ranges of MT and DT HOM, the change in transmission is likely less, since the relative change in ion mass is smaller. This behavior is shown e.g. by Heinritzi et al. (2016). Nevertheless, we only give a very rough estimate of HOM concentration in this study, with a large uncertainty given as "at least +100%/-50%"which to a large part is due to the uncertainty in the transmission of the instrument. We also emphasize in the manuscript (Lines 255-258) that the transmission is the most likely reason for an even larger error in the concentration estimate.

To put it differently, we only estimate the calibration factor based on previous studies, and it would not be motivated to change this value based on an unknown transmission. It is only the estimated uncertainty that we can adjust to reflect potential problems related to the transmission, or any other undesired effects or parameters.

5. Ent-kaurene concentrations in this paper are at the ppb level, while in real atmosphere ppt or ppq levels. The reactant concentration obviously has an effect on RO2 chemistry. Then is the HOM yield determined in this paper applicable to the real atmosphere? The authors would need to discuss this.

Response: To address this issue, we added a paragraph discussing the atmospheric relevance of the estimated HOM yield on Page 12. We believe the HOM yield determined in our study is applicable to the real atmosphere for two reasons. On the one hand, the ent-kaurene concentrations in this study were indeed higher than those monitored in the atmosphere, but the ~2 % HOM yield we report is based on periods with kaurene concentrations lower than 50 ppt (Fig. S3(b)). On the other hand, in this study, $RO_2$ is only formed via kaurene oxidation, while in the real atmosphere, and there are also other abundant radicals, for example, $HO_2$, NO, and $RO_2$ from other pathways, all of which can impact $RO_2$ chemistry and HOM formation.

*"HOM yields from kaurene ozonolysis, as for any other system, will always depend on the atmospheric conditions, in particular the type and concentration of bimolecular reaction partners (mainly NO, $RO_2$, $HO_2$). The reaction partner will determine the possible branching pathways, while their absolute concentrations will affect $RO_2$ lifetimes, and thereby the potential to undergo autoxidation. According to previous studies, the reported kaurene concentrations*

*in the atmosphere range from ppq to a few ppt (Li et al., 2020; Yee et al., 2018). In our study, concentrations were higher (Fig. S3(a) and (c)), but the ~2 % HOM yield was estimated based on periods when kaurene concentrations were < 50 ppt (Fig. S3(b)).As the atmosphere is much more complex, with various other species present, it is possible that the $RO_2$ concentrations in our experiments are, in fact, lower than in atmosphere, despite the kaurene concentrations being higher. Nevertheless, we think that the oxidation conditions in our experiments are quite close to atmospheric conditions, and that the potential bias in HOM yields due to this is marginal when compared to the overall uncertainty of HOM quantification (section 2.4.2). There are two main exceptions to this: firstly, if NO concentrations would be very high, it could lead to very different oxidation pathways, with unknown effects on the HOM yields. Secondly, in the atmosphere it would be much less likely for two kaurene-derived $RO_2$ to react, as the majority of the $RO_2$ would come from different precursors, and thus the $C_{37-40}$ dimers would be unlikely to form in the atmosphere."*

[Figure]

Figure S3. HOM yield estimation and kaurene concentrations. HOM concentrations were plotted against the kaurene ozonolysis rate in panels (a) and (b). Markers are colored by kaurene concentrations. Lines are added to the plots to represent constant HOM molar yields from kaurene ozonolysis, accounting for an instrumental HOM background in the CI-APi-TOF of 0.04 ppt. (a) All data points (plotted in logarithmic scale) from the periods without seed particles in the chamber. A zoomed-in view of panel (a) with reaction rates ranging from 0 to 0.05 ppt s$^{-1}$ is shown in panel (b) (plotted in linear scale). Panel (c) shows the kaurene levels during the periods shown in panel (a).

6. The authors would need to clarify how the uncertainty for HOM yield is obtained, i.e. show the calculation of error propagation.

Response: Thanks for pointing this out. We realized that we had only discussed uncertainties related to HOM concentrations but not the yields. The obvious challenge is that we can only estimate the uncertainties, as the error sources are so diverse, but we have now included a rough estimation on the yield uncertainty at the end of section 2.5 on Page 9 (Lines 281-284):

*"The HOM quantification alone had a large uncertainty, which we can only estimate to be at least +100%/-50%. Also the kaurene quantification required several assumptions, as did $k_1$. As for HOM quantification, we can only make estimations on the uncertainties of these parameters. If we assume that these three are the major error sources, and each of them have the same uncertainty of +100%/-50%, the final uncertainty (via error propagation) becomes +173%/-87%."*

7. As to product volatility, I am not sure if varying the condensation sink is the right way to probe it. First of all, increasing the CS would reduce the lifetime of some first-gen products, and in turn alter the second-gen chemistry, which may influence the fraction remaining. Moreover, I think it is the accommodation coefficient and not the volatility that faction remaining associates to. This is probably why LVOC and ELVOC have pretty much the same faction remaining (their accommodation coefficients are probably all near unity), while their volatilities differ by several orders of magnitude. The authors would need to re-think and rephrase the related discussion.

Response: We had indeed not described the approach in enough details in the manuscript, and this has now been changed. We also more clearly included references to some of our earlier studies where we have used the same approach, and discussed the method in much more detail, in particular Peräkylä et al. (2020). By comparing the experiment with modeled results, they found the fraction remaining of different oxidation products after seed injections was a function of the volatility of the compounds, though obviously not with any direct linear dependence, but rather as a logistic function (Peräkylä et al., 2020).The accommodation coefficient is also closely related both to the fraction remaining, and to the volatility. We have now included also this in the discussion.

We expect the condensation to seed particles to be the main cause of HOM changes during the experiment, as we consider oxidation products originating directly from kaurene oxidation to be the main HOM source. Second generation products would primarily be formed from the OH oxidation of first generation oxidation products, but the OH concentration is going to be low, as is the likelihood of OH to react with these products, considering the competition from kaurene and the wall contaminants that were observed to form accretion products. $RO_2$ intermediates themselves have short lifetimes, and the condensation of $RO_2$ on aerosol particle should be a smaller sink than their bimolecular reactions. During a typical seed injection, the drop of gas-phase concentration of $RO_2$ are estimated to be less than 10%. This would in turn decrease the source of all closed shell oxidation products formed in the reactions.

We have made the following additions to the text, on Lines 389-395, Page 13:

*"This method has been applied to study the volatilities of the α-pinene and cyclohexene oxidation products by Peräkylä et al. (2020) and Räty et al. (2021). In this methodology, we assume the condensation to seed particles is the main driver of the concentration changes of oxidation products during the experiment, although other changes (e.g., the changes of precursors and intermediates due to seed injection) can have a non-negligible influence (Peräkylä et al., 2020). This approach does not provide direct estimates of volatilities, but it does provide a good*

*separation between readily condensing vapors (accommodation coefficient near unity) and the more volatile ones that are unaffected by the condensation sink (i.e., with very low accommodation coefficients)."*

Minor comments:

1. Line 27: "with 4–5 O-atoms needed for the main monomeric species to condense". It might make sense to modify the conclusion by adding constraints like "to condense onto xx nm particle". (In theory, any compound can condense at sufficient high supersaturation.)

   Response: We have revised the sentence on Line 28, Page 1 accordingly, though instead saying "under our experimental conditions". As the reviewer points out, anything will condense at suffiently high super saturation, and in that case particle size is not the critical parameter.

2. Line 40: remove "new".

   Response: Removed.

3. Line 126: change "after a collision" to "after collisions".

   Response: Changed.

4. Line 280: move the experiment conditions to the "Methods" section.

   Response: Moved.

5. Line 334: "It is also possible that the nitrate CI-APi-TOF might be more sensitive towards less oxygenated dimers if there are 40 C-atoms rather than 20 in their skeleton, if a larger molecule increases the likelihood of forming a stable cluster with NO3-". Is it possible? Please explain more.

   Response: The detection of a molecule in the CI-APi-TOF depends on the stability of the cluster formed with the reagent ion. The more oxygenated HOM typically have suitable functionalities that can form several hydrogen bonds with $NO_3^-$, and are thus detected with high sensitivity. For smaller molecuels, the positions of these groups need to be suitable, and here our speculation is based on a larger molecule having more conformations in which it can form suitable bonds with the $NO_3^-$.

6. Line 468: should the "with carbon skeleton" be removed?

   Response: Removed.

7. The authors state in line 365: "HOM yield ranged between 0.1%-3%", while in Table 1 "0.5-10", which one is correct

Response: "HOM yield ranged between 0.1%-3%" should be "HOM yield ranged between 0.1%-10%" (we have revised this mistake on Line 367, Page 12). And this range was obtained based on the whole dataset (experiments both at low and high oxidation rates). In Table 1, we only report the HOM yield estimated at low oxidation rates (kaurene concentrations were below 50 ppt which was more close to the real atmospheric level), and "0.5-10" (we have revised it as 0.27-5.5 based on the newly propagated uncertainty in HOM yield) was the range considering the uncertainties.

**RC2**

The authors describe experimental results from the ozonolysis of the diterpene kaurene (C20H32) carried out in a low-flow 2 m3 Teflon reactor for close to atmospheric conditions and RH < 1% with a residence time of the reaction mixture of about 1 hour. Kaurene was dosed by flushing a carrier gas through a heated vial filled with the hydrocarbon and introducing the flow into the reactor. Gas-phase analysis was carried out applying a Vocus PTR mass spec and a nitrate-based CIMS. A SMPS system and a AMS analyzed the formed particles. Resulting kaurene concentrations in the Teflon reactor were tried to determine using Tenax absorbers and by its O3 reactivity with and assumed rate coefficient k(O3+kaurene). Generally, this manuscript describes a very challenging experimental work. A well-defined conversion of such a sticky compound in this reactor type is a really hard job. This manuscript represents the first reported product study of a diterpene. Despite the big uncertainty of stated product concentrations and formation yields, the results are interesting and worth to be published in ACP. Some points should be considered before final acceptance is recommended.

1. Line 96: What is the reason using RH <1%?

   Response: As seen from our manuscript, and the reviewer also notes, the experiments were very challenging. We conducted our experiments under dry conditions to eliminate any additional complicating influence from water. While obviously not mimicking the atmosphere in this regard, the gas-phase oxidation chemistry is typically not strongly impacted by RH. The SOA formation can be sensitive to the humidity, but this was not studied in this manuscript. If we have a humid chamber, heterogeneous reactions may be facilitated and also the effective condensation sink can be altered from the SMPS-derived values (as the SMPS always measures dry diameters).

2. Line 99: Is something known in the literature regarding vapor pressure and melting/boiling point of kaurene?

   Response: Kaurene used in this study was purchased from OIChemIm s.r.o, and the melting point given in the instruction is 50-51 ℃. No information regarding vapor pressure and melting/boiling point of ent-kaurene was found in literature (experiment data). However, the predicted information is listed as follows:

|  | Predicted values I[*] | Predicted values II[#] |
|---|---|---|
| melting point (℃) | 97.37 | |
| boiling point (℃) | 316.2 | 346.9 at 760 mm Hg |
| vapour pressure at 25 ℃ (mm Hg) | 0.000314 | $0 \pm 0.4$ |

*predicted data is generated using the US Environmental Protection Agency's EPISuite.

**predicted data is generated using the ACD/Labs Percepta Platform - PhysChem Module.**

During our experiments, the kaurene did melt when we heated to the highest temperatures used, which we estimate to be around 60 degrees, suggesting that the values by the manufacturer are more accurate than the predicted values.

3. Line 155: Poor reliability of kaurene quantification is a weakness of this work. Did the authors think about a GC-FID technique applying the "effective carbon number" approach, which is especially suitable for pure hydrocarbons with a relatively high carbon number. It´s an old, but very robust, technique for measuring concentrations of hydrocarbons as successfully used for measuring sesquiterpenes in kinetic and product studies for atmospheric conditions. Should work for kaurene as well.

Response: A GC-FID would have been a valuable asset in the quantification. However, we did not have such an instrument available, and we expected that the combination of multiphase adsorbent (Tenax) tube sampling (quantified offline using TD-GC-MS) and the Vocus PTR would have provided a good enough quantification. The main reason for the large uncertainty in kaurene quantification was the unfortunate blockage of the pinhole in the Vocus, which was only realized at the end of the measurements. If the Vocus would have worked as expected during the experiment, the sensitivity to kaurene would have been identical during the whole experiment, and a single calibration factor could have been used. The effects on ozone when adding kaurene would then have worked as a validation, while now it largely became the basis for the quantification. This is very regrettable, but cannot be helped at this stage.

4. Line 183: I think the chosen rate coefficient k(O3+kaurene) = 5 x 10-15 cc/s cannot be estimated from modeling as described in the SM. There are 4 free parameters (kaurene feed, k(O3+kaurene), k(OH+kaurene) and OH yield) you can play with. There is not enough experimental information to fix them independent from each other. Is k(O3+kaurene) available via SAR? What about to measure k(O3+kaurene) using a relative rate technique? A signal, proportional to the kaurene concentration, is available from the PTR-MS. With the knowledge of a more reliable k(O3+kaurene) (and in the presence of an OH scavenger), a better estimate of the kaurene concentration should be possible based on the measured O3 disappearance.

Response: We do not agree that the $k(O_3+kaurene)$ cannot be estimated using the approach we present. It is, of course, strictly an estimate, but based on the results in Fig. S1, it is clear that there is a limited range of values that can be used to match observations. The modeled kaurene concentrations at $k(O_3+kaurene) = 5\times10^{-16} cm^3 s^{-1}$ are the closest to the measured in all simulated cases, and if using a value of $1\times10^{-16} cm^3 molec^{-1} s^{-1}$ or less, or a value of $1\times10^{-15} cm^3 s^{-1}$ or higher, it is impossible to capture the relative change in measured kaurene trace. While there are indeed four free parameters, Fig. S1 also shows that the relative change in kaurene or ozone is not very sensitive to k(OH+kaurene) and OH yield.

There are only two k(O$_3$+kaurene) available in previous study. An experimental value of $1.4 \pm 8.2 \times 10^{-15}$ cm$^3$s$^{-1}$ has beem reported only by Helin et al. (2020), while the value estimated by EPISuite$^{TM}$ (US EPA) based on SAR is $1.2 \times 10^{-17}$ cm$^3$s$^{-1}$. The significant difference between these two estimated values (two orders of magnitude), and the inability of our kinetic model to match the observations with either value, motivated us to re-estimate the k(O$_3$+kaurene). We tested k(O$_3$+kaurene) within the range of $1 \times 10^{-15} - 1 \times 10^{-17}$ cm$^3$s$^{-1}$ covering the previously reported values (we only show the results of k= $1 \times 10^{-15} - 1 \times 10^{-16}$ cm$^3$s$^{-1}$ because the modeled O$_3$ and kaurene concentrations at k = $5 \times 10^{-17}$ and $1 \times 10^{-17}$ were too far from the measured ones). Our estimation (k(O$_3$+kaurene) = $5 \times 10^{-16}$ cm$^3$s$^{-1}$) was within the uncertainties of the value reported by Helin et al. (2020). Thus, we are convinced that the k(O$_3$+kaurene) given in our study is a reasonable estimate.

We agree that the estimation of the reaction rate coefficient would have been easier in the presence of an OH scavenger. However, the period with CO (OH scavenger) injection in our experiment was not suitable for a similar in-depth analysis model to estimate k(O$_3$+kaurene) as in Fig. S1. This is because during this period, instead of injecting ozone into the chamber containing kaurene, the kaurene was added into the O$_3$-stable chamber. In this case we can no longer use the relative change in kaurene concentration, and we are not left with enough constraints. However, as shown below, when adding kaurene during the CO experiments, our model does nicely match the changes in ozone when using k(O$_3$+kaurene) = $5 \times 10^{-16}$ cm$^3$s$^{-1}$ and the kaurene concentration we derived.

[Figure]

*The whole period shown in this figure was with the presence of CO. The kaurene concentrations was used as the input in the model and OH reactions were turned off. The modeled ozone change at different k(O$_3$+kaurene) are shown with different red lines. As the difference in the rapidity of the ozone change for cases, the steady-state value during ~17:50 – 19:00 which we focus on.

5. Line 230: A reaction scheme showing the expected first pathways would be helpful to better understand the argumentation given in this paragraph.

Response: We have added a reaction scheme to clarify the standard mechanisms leading to the primary RO₂ from kaurene-ozone and kaurene-OH reactions in Figure 3 (Page 24). We do not explicitly draw out any following reactions leading to closed-shell species, as our data cannot provide support for any particular steps among the many possibilities that exist.

[Figure]

Figure 3. Simplified mechanism of the formation routes of the primary $RO_2$ from kaurene-$O_3$ and kaurene-OH reactions. POZ: cyclic primary ozonide; CI: Criegee intermediates; SCI: stabilized Criegee intermediates; VHP: vinyl hydroperoxides.

6. Line 252: Is the "calibration factor" $c = 10^{10}$ molecules/cc not coming from sulfuric acid calibration? Is there an uncertainty only with a factor of 2? Was duty cycle correction applied? Additional uncertainty arising from the ion transmission of the big ions is mentioned. What is the expected total error in HOM concentrations?

Response: Yes, the calibration factor C=1×10$^{10}$ cm$^{-3}$ comes from sulfuric acid calibration in most of the cited works. We didn't apply a duty cycle correction, as we consider the uncertainty arising from the ion transmission to be larger than (and at the largest masses opposite to) the duty cycle effect. We also refer to our response to reviewer 1 (Comment 4), and stress that we state the uncertainty as "at least" a factor of 2.

7. Line 273: Also here, a (rough) estimate of the uncertainty in the HOM yield should be given.

Response: Thank you for this suggestion. We have now included a rough estimation on the yield uncertainty at the end of section 2.5 on Page 9 (Lines 281-284):

*"The HOM quantification alone had a large uncertainty, which we can only estimate to be at least +100%/-50%. Also the kaurene quantification required several assumptions, as did $k_1$. As for HOM quantification, we can only make estimations on the uncertainties of these parameters. If we assume that these three are the major error sources, and each of them have the same uncertainty of +100%/-50%, the final uncertainty (via error propagation) becomes +173%/-87%."*

**References:**

US EPA. Estimation Programs Interface Suite™ for Microsoft® Windows, v 4.11, United States Environmental Protection Agency, Washington, DC, USA, 2021.

Heinritzi, M., Simon, M., Steiner, G., Wagner, A. C., Kürten, A., Hansel, A., and Curtius, J.: Characterization of the mass-dependent transmission efficiency of a CIMS, Atmos. Meas. Tech., 9, 1449-1460, 10.5194/amt-9-1449-2016, 2016.

Helin, A., Hakola, H., and Hellén, H.: Optimisation of a thermal desorption–gas chromatography–mass spectrometry method for the analysis of monoterpenes, sesquiterpenes and diterpenes, Atmospheric Measurement Techniques 13, 3543-3560, 10.5194/amt-13-3543-2020, 2020.

Li, H., Riva, M., Rantala, P., Heikkinen, L., Daellenbach, K., Krechmer, J. E., Flaud, P. M., Worsnop, D., Kulmala, M., Villenave, E., Perraudin, E., Ehn, M., and Bianchi, F.: Terpenes and their oxidation products in the French Landes forest: insights from Vocus PTR-TOF measurements, Atmospheric Chemistry and Physics, 20, 1941-1959, 10.5194/acp-20-1941-2020, 2020.

Peräkylä, O., Riva, M., Heikkinen, L., Quéléver, L., Roldin, P., and Ehn, M.: Experimental investigation into the volatilities of highly oxygenated organic molecules (HOMs), Atmospheric Chemistry and Physics, 20, 649-669, 10.5194/acp-20-649-2020, 2020.

Räty, M., Peräkylä, O., Riva, M., Quéléver, L., Garmash, O., Rissanen, M., and Ehn, M.: Measurement report: Effects of NOx and seed aerosol on highly oxygenated organic molecules (HOMs) from cyclohexene ozonolysis, Atmospheric Chemistry and Physics, 21, 7357-7372, 10.5194/acp-21-7357-2021, 2021.

Yee, L. D., Isaacman-VanWertz, G., Wernis, R. A., Meng, M., Rivera, V., Kreisberg, N. M., Hering, S. V., Bering, M. S., Glasius, M., and Upshur, M. A.: Observations of sesquiterpenes and their oxidation products in central Amazonia during the wet and dry seasons, Atmospheric Chemistry and Physics, 18, 10433-10457, 10.5194/acp-18-10433-2018, 2018.